# Prioritizing Sustainable City Indicators for Cambodia

**Puthearath Chan [1,2,](ORCID) and Myeong-Hun Lee [2]**

[1] Office of Sustainable Lifestyle, Ministry of Environment, Phnom Penh 12301, Cambodia
[2] Graduate School of Urban Studies, Hanyang University, Seoul 04763, Korea; mhlee99@hanyang.ac.kr
[*] Correspondence: cptr5@hanyang.ac.kr or puthearath_chan@moe.gov.kh

**Abstract:** This research is based on our previous research that developed consensus sustainable city indicators for Cambodia through three-round Delphi panel surveys. That research developed indicators in the first round based on UN sustainable development goal 11, ASEAN environmentally sustainable city, Korean case study, and domestic green and clean city indicators, and validated the developed indicators in the last two rounds. After consensus analysis, that research obtained 32 assessment indicators categorized by nine criteria. However, these indicators are not prioritized yet due to the limitation of the Delphi technique. Hence, this research aims to prioritize these indicators by applying the Analytic Hierarchy Process (AHP) technique and to confirm whether the levels of importance verified by Delphi can be used for prioritizing or ranking the indicators. This research surveyed potential respondents experienced and working in relevant fields both offline and online. Online surveys were processed through E-mail, Facebook, and LinkedIn. A total of 118 questionnaires were gathered from the surveys, and 16 were inconsistent (consistency ratio > 0.1). The results showed that the highest and lowest weights are 0.0557 and 0.086. The top ten indicators are slum population (0.0557), unemployment (0.0516), crime prevention (0.0470), water supply (0.0469), city's migration (0.0462), low-income housing (0.0445), solid waste collection (0.0437), labor-force (0.0421), construction safety (0.0400), and traffic congestion (0.0398). The rank of all indicators based on their levels of importance is completely different from the rank of their weights. Therefore, this research confirms that the levels of importance verified by Delphi cannot be used for ranking or prioritizing the consensus indicators. The priority weights in this research would be useful to policymaking, strategic direction, and budget allocation for the development and management of sustainable cities in Cambodia.

**Keywords:** Delphi panel survey; AHP priority calculator; Cambodia sustainable city assessment; UN sustainable development goals; sustainable urban development; priority weight analysis

## 1. Introduction

### 1.1. Background and Purposes

Assessment indicators for the development and management of sustainable cities in Cambodia were developed by a previous research [1] through the processes of Delphi panel surveys. The Delphi technique was chosen to develop and validate the indicators in that research because this technique, as illustrated in many studies [2–9], is suitable to (a) obtain accurate information that is unavailable, (b) handle complex problems that require more judgmental analysis, (c) define areas where there is considerable uncertainty and/or a lack of agreed knowledge or disagreement, (d) allow for combining fragmentary perspectives into a collective understanding, (e) model a real world phenomena involving a range of viewpoints and for which there is little established quantitative evidence, and (f) highlight topics of concern and assess uncertainty in a quantitative manner.

That Delphi research was carried out through three-round panel surveys. The first round was to develop the indicators. The second round was to pre-validate the indicators (to identify the levels of importance). The third round was to validate the indicators (to confirm the levels of importance). The first-round questionnaire was developed based on the five major source indicators such as UN sustainable development goal 11, ASEAN environmentally sustainable city, Korean HAN case study, and domestic green and clean city indicators (see Appendix A, Table A1). The UN sustainable development goal 11 (SDG 11) indicators resulted from the major step forward and an improvement on the millennium development goals (MDGs) [10,11] and agreed in the UN 2030 agenda for sustainable development. The sustainable development goals addressed 17 goals and 169 targets [12–15]. The goal 11 addressed 10 targets, and its indicators were reviewed in the column 'SDG 11' [16,17]. The ASEAN environmentally sustainable city (ESC) indicators were developed by the Association of Southeast Asian Nations (Brunei, Cambodia, Indonesia, Laos, Malaysia, Myanmar, Philippines, Singapore, Thailand, and Vietnam) which was endorsed by ASEAN Environment Ministers in 2005. The goal is to pursue environmental sustainability in the rapidly growing cities of ASEAN countries [18]. The ASEAN ESC indicators were reviewed in the column 'ESC' [19]. The Korean HAN indicators refer to the Korean case study indicators developed by HAN Sang Mi [20]. This case study developed indicators based on the UN SDG 11 indicators, HABITAT indicators, and Korea's relevant indicators. These indicators were reviewed in the column 'HAN' [21]. The green city indicators refer to the indicators developed under the green city development project in Cambodia. The Cambodia National Council for Sustainable Development (NCSD) implemented the project named "Green Urban Development Program" with the support from the Global Green Growth Institute (GGGI) and produced the green city strategic planning methodology [22] as well as green city strategic plan for Phnom Penh 2017 to 2026 [23]. The proposed indicators attached with their sectoral objectives were reviewed in the column 'GC'. The clean city indicators refer to the indicators addressed in the clean city standard of Cambodia. This standard was developed by the Cambodia National Committee for Clean City Assessment (NCCCA) aiming to monitor and assess Cambodia's cities through the clean city contest every three years. The winning cities will be awarded by the Prime Minister of Cambodia in the following three names "Clean City Romduol I, II, and III" upon the winning score [24,25]. The clean city indicators were reviewed in the column 'CC' [26].

The above Delphi research used these indicators to identify relative categories for developing the questionnaire for round one, especially to supplement the measurement-lacked indicators obtained from the round one. As a result, 69 initial indicators were obtained from the round-one survey. These 69 indicators were brought into the validation process. By using a 5-point Likert-type scale, the 69 indicators were reduced to 41 indicators in the first validation through the round-two survey. By using both the 5-point Likert-type scale and mean values (levels of importance) obtained from round two, the 41 indicators were reduced to 32 indicators in the final validation through the round-three survey. After validating the developed indicators, the research accordingly analyzed the consensus. Finally, the research obtained 32 consensus indicators categorized by nine criteria.

However, the above research does not prioritize these consensus indicators yet due to the limitation of the Delphi technique. As priority weight is necessary for the sectoral and inclusive assessment of sustainable cities [1,20], this research aims to prioritize these indicators. Furthermore, many studies showed that the Analytic Hierarchy Process (AHP) technique can be used for prioritizing issues such as policies, criteria, index, and indicators [27–31]. In particular, its special characteristic "Pairwise comparison" is very significant in the prioritization known as weight analysis [32–35]. Therefore, this research will apply the AHP technique for prioritizing the 32 consensus indicators, including the nine criteria and the assessment indicators in each criterion. Moreover, as the previous research verified the levels of importance of the consensus indicators by using the Delphi technique, this research also aims to address the question "How is the rank of the consensus indicators based on levels of importance (Delphi) compared to their rank based on relative weights (AHP)?" in order to confirm whether the levels of importance verified by Delphi can be used for ranking or prioritizing the consensus indicators. The hypothesis of this research is "AHP's rank is different from the Delphi's rank".

*1.2. Consensus Indicators*

There are 32 consensus indicators developed by the previous research [1]. These indicators are categorized by nine criteria. The nine criteria are (1) Demography, (2) Employment, (3) Housing, (4) Transport, (5) Safety, (6) Water Use, (7) Waste Management, (8) Air Quality and Energy, and (9) Urban Spaces and Tourism. These criteria and their assessment indicators are shown in Table 1.

**Table 1.** The nine criteria and their assessment indicators.

| Criteria | Indicator | Description |
|---|---|---|
| Demography | Population density | This indicator assesses the living spaces in the city. It measures through the density of population per square kilometer. |
| | City's migration | This indicator assesses the migration situation in the city. It measures through the rural-urban migration rate in a year. |
| | Household income | This indicator assesses the economic conditions of households living in the city. It measures through the income of households. |
| Employment | Labor forces | This indicator assesses the productivity of the city. It measures through the productive population in the labor forces. |
| | Unemployment | This indicator assesses the employment situation in the city. It measures through the unemployment rate. |
| | New jobs creation | This indicator assesses the city government's efforts in creating new jobs towards reducing the unemployment rate. It measures through the number, type, and size of new jobs created per year. |
| Housing | Slum population | This indicator assesses the residential and living environment in the city. It measures through a percentage of the population living in slums or informal/unplanned settlements. |
| | Low-income housing | This indicator assesses the city government's efforts in providing affordable housing. It measures through the number, type, and size of the low-income housing development projects. |
| | Quality of buildings | This indicator assesses the quality of buildings in the city focused on residential buildings. It measures through the percentage of the new residential buildings (aged less than 30 years). |
| Transport | Transport means | This indicator assesses the public transport sharing rate in the city. It measures through a percentage of public transport means compared to total transport means in the city. |
| | Sidewalks | This indicator assesses the city government's efforts in improving sidewalks for pedestrians. It measures through the number, type, size of initiated programs or activities for improving sidewalks. |
| | Parking lots | This indicator assesses the parking situation in the city, including the city's government efforts in improving public parking lots. It measures through the number, type, and size of parking lots and improvement initiatives of the city government. |
| | Traffic congestion | This indicator assesses the city government's efforts in reducing traffic congestion. It measures through the number, type, and size of initiated programs or activities for reducing traffic congestion. |
| Safety | Crime prevention | This indicator assesses the city government's efforts in preventing crimes in the city. It measures through the number and type of measures or initiatives of the city government to prevent crimes. |
| | Construction safety | This indicator assesses the city government's efforts in preventing construction risks. It measures through the number and type of initiated programs or activities to prevent construction risks. |
| | Disaster prevention | This indicator assesses the prevention facilities of the city's government to prevent disasters. It measures through the number, type, and size of existing disaster prevention facilities. |
| | Insurances | This indicator assesses the social welfare situation in the city. It measures through a percentage of the population registered in the insurance system compared to the total population in the city. |

**Table 1.** *Cont.*

| Criteria | Indicator | Description |
|---|---|---|
| Water Use | Water supply | This indicator assesses the situation of water supply in the city. It measures through a percentage of households with access to potable water supply infrastructure compared to total households. |
| | Water consumption | This indicator assesses the situation of water consumption in the city. It measures through an average amount of water consumed by a person or household daily, weekly, or monthly. |
| | Water reservoirs | This indicator assesses the situation of freshwater supply sources in the city. It measures through the number, type, and size of natural or artificial reservoirs in or nearby the city. |
| Waste Management | Solid waste collection | This indicator assesses the public organizations for solid waste collection and the city government's efforts in improvement. It measures through a percentage of households linked to collecting network and number and type of improvement initiatives. |
| | Wastewater treatment | This indicator assesses the situation of wastewater treatment in the city. It measures through the number, type, and size of wastewater treatment plants used for treating wastewater in the city. |
| | Waste reduction | This indicator assesses the city government's efforts in reducing wastes. It measures through the number, type, and size of initiated programs, activities, or measures to reduce waste in the city. |
| Air Quality and Energy | Fine dust levels | This indicator assesses the air quality in the city, including the city government's efforts in reducing greenhouse gases (GHGs). It measures through PM levels and number, type, and size of initiated programs, activities, or measures to reduce GHGs. |
| | Urban forest | This indicator assesses the forest cover in the city, including the city government's efforts in planting trees. It measures through a percentage of total forest cover in the city, and number, type, and size of initiated programs or activities to plant more trees. |
| | Energy consumption | This indicator assesses the situation of energy consumption in the city. It measures through an average amount of electricity consumed by a person or household and number of initiated programs or activities for energy saving in daily life. |
| | Renewable energy | This indicator assesses the renewable energy production and the city's government efforts for promoting renewable energy. It measures through a percentage of renewable energy contributed to the electricity supply and number, type, and size of the initiatives for promoting renewable energy use and production. |
| Urban Spaces and Tourism | Urban parks | This indicator assesses the public green spaces or parks in the city. It measures through the number and size of natural or artificial parks in the city, including their accessibility and cleanliness. |
| | Botanic gardens | This indicator assesses the biodiversity gardens created for tourism, education, and conservation purposes in the city. It measures through the number, type, and size of the gardens. |
| | Heritage conservation | This indicator assesses the conservative situation of cultural, historical, and heritage status or buildings in the city. It measures through the numbers of the conserved status or buildings, including budgets allocated to preserve that status or buildings. |
| | Tourism growth | This indicator assesses the tourists' satisfaction for the city, including the city's government efforts in attracting more tourists. It measures through tourism growth rate and satisfaction level and the programs or activities initiated to attract more tourists. |
| | Playgrounds | This indicator assesses the situation of playgrounds or leisure areas in the city. It measures through the number, type, and size of the playgrounds and leisure areas in the city. |

## 2. Materials and Methods

As mentioned in Section 1.1, this research applied the Analytic Hierarchy Process (AHP) technique to prioritize the nine criteria and their assessment indicators. These prioritizations are divided into three parts. The first part is the prioritization of the nine criteria. The second part is the prioritization of the assessment indicators in each criterion. The third part is the prioritization of the 32 assessment indicators from all criteria. As shown in Figure 1, the nine criteria and their number of assessment indicators for AHP surveys are Demography (three indicators), Employment (three indicators), Housing (three indicators), Transport (four indicators), Safety (four indicators), Water Use (three indicators), Waste Management (three indicators), Air Quality and Energy (three indicators), and Urban Space and Tourism (five indicators).

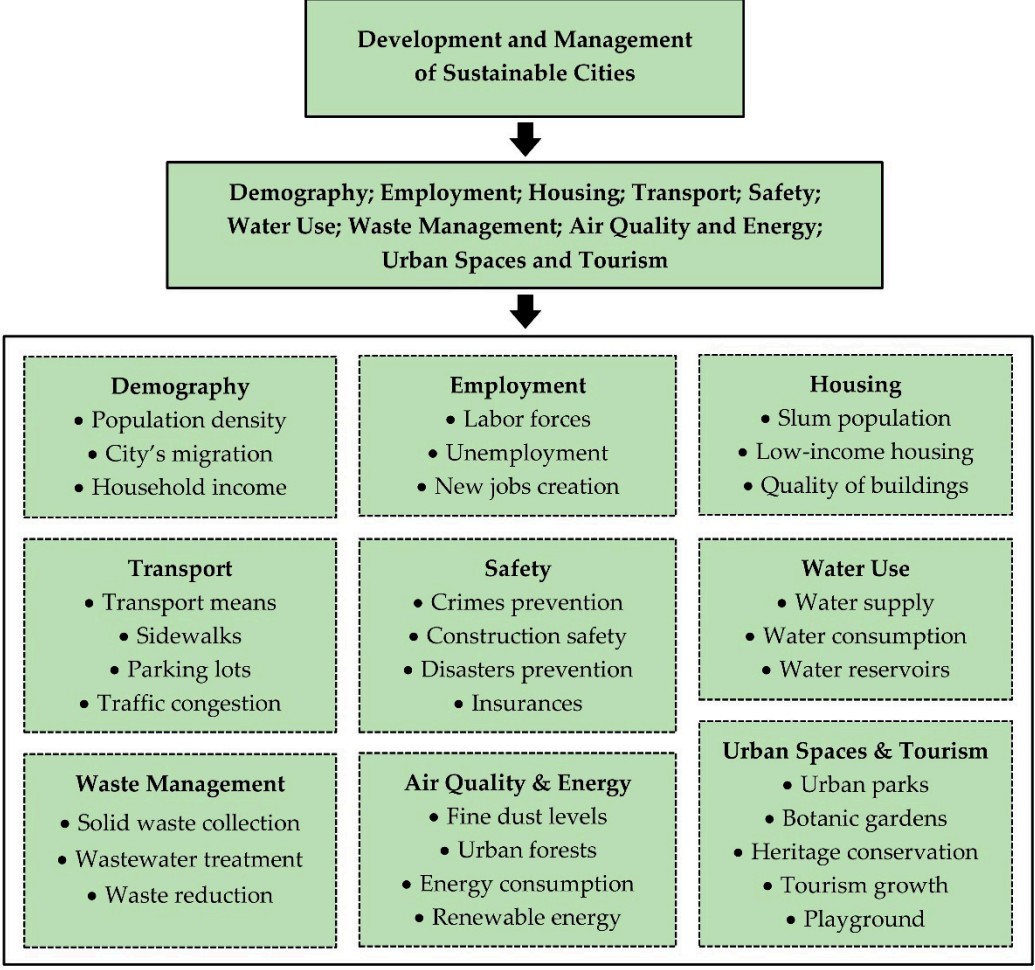

**Figure 1.** The nine criteria and their assessment indicators for AHP surveys.

### 2.1. Questionnaire Development

There are two parts of the AHP questionnaire for the survey of this research. The first part is the AHP questionnaire for prioritizing the nine criteria. The sample of this questionnaire is shown in Figure A1. The second part is the AHP questionnaire for prioritizing the assessment indicators in each criterion. The sample of this questionnaire is shown in Figures A2–A4. Furthermore, the AHP questionnaire commonly uses the scale from 1 to 9 for its pairwise comparisons which is 1 "Equal importance", 3 "Moderate importance", 5 "Strong importance", 7 "Very strong importance", 9 "Extreme importance", and 2,4,6,8 "Values in-between" [36]. Moreover, the questionnaire for prioritizing the nine criteria started with the question "Between the Criteria **A** and **B**, which one is more important

for the assessment of sustainable city development and management in Cambodia?" Likewise, the questionnaire for prioritizing the assessment indicators in each criterion started with the question "Between the Indicator **A** and **B**, which one is more important for the assessment of sustainable city development and management in Cambodia?" These questions and their questionnaires are shown in Appendix B. Especially, all of the 32 indicators and their explanations were translated to Khmer language, and both Khmer and English versions including the published previous research paper were all attached with questionnaires sent to the targeted respondents.

### 2.2. AHP Survey and Data Analysis

AHP technique uses pairwise comparison for prioritizing the criteria and/or indicators wildly known as weight analysis [37–39]. This technique is very popular in qualitative assessment and used in various research, including the research theses/dissertations of the graduate students [40–45]. For inclusive assessment research, especially the research on the topics of sustainable cities, this technique requires the number of respondents to be at least 100 [20,21,46,47]. Moreover, the previous research suggested prioritizing the criteria and their assessment indicators with this sample size as well. The previous research further suggested that the respondents must be experienced and/or working in the fields related to clean, green, and sustainable city development and management, especially related to the nine categories (criteria) of the consensus indicators [1]. By following these studies, this research determined that the sample size for this next-step research must be at least 100 respondents who are working and experienced in the above-mentioned relevant fields. In this case, this research selected the relevant respondents from various institutions, such as governmental institutions, non-governmental organizations (NGOs), private firms/companies, and others (research institutions, independent researchers, research students, experienced retirees, and activists). Table 2 shows the number of respondents for both distributed and gathered surveys, including consistent (valid) and inconsistent samples classified by the types of workplaces and survey methods "Paper, Email, Facebook, and LinkedIn". These offline- and online-survey methods were used to collect data from all relevant and potential respondents in order to obtain the valid sample size of "At least 100".

In fact, the pairwise comparisons through the AHP technique must be consistent [48,49]; thus, this research selected only consistent samples. The consistency ratio must be less than or equal to 0.1 (CR ≤ 0.1 or CR ≤ 10%) [36,50–52]. Therefore, the gathered questionnaires were analyzed from day to day during the time of the survey and sometimes immediately after the survey. This analysis used both Microsoft Excel and AHP-OS online program. The AHP-OS program (Figure 2) was developed by Prof. Dr. Klaus D. Goepel [36]. This program is accessible in [53], and its AHP Priority Calculator is accessible in [54]. This priority-calculator program is also available to download the data in Microsoft Excel easily and immediately after the analysis. The explanations of this program were published in [36,55]. Although this program is very accessible and easy to use, this research still used Microsoft Excel mostly. This was because the internet connection was limited during the mission to collect the data through the offline survey in Cambodia. It means that the offline surveys "Paper: Face-to-face interviews" have been conducted during the research mission of P.C. to Cambodia from 13 July 2019 to 11 August 2019. And the online surveys were started from 5 June 2019 until obtaining the consistent samples of 102 on 2 September 2019.

**Table 2.** Statistics of the survey.

| Survey Methods | Institutions of Respondents | Questionnaires | | | |
|---|---|---|---|---|---|
| | | Distributed | Gathered | Inconsistent | Consistent |
| Paper | Government | 25 | 25 | 1 | 24 |
| | NGOs | 8 | 8 | 0 | 8 |
| | Private Firms | 2 | 2 | 0 | 2 |
| | Others | 4 | 4 | 1 | 3 |
| | **Total I** | **39** | **39** | **2** | **37** |
| Email | Government | 23 | 13 | 4 | 9 |
| | NGOs | 24 | 16 | 5 | 11 |
| | Private Firms | 19 | 5 | 0 | 5 |
| | Others | 12 | 3 | 1 | 2 |
| | **Total II** | **78** | **37** | **10** | **27** |
| Facebook | Government | 10 | 10 | 0 | 10 |
| | NGOs | 3 | 3 | 0 | 3 |
| | Private Firms | 10 | 10 | 2 | 8 |
| | Others | 8 | 8 | 1 | 7 |
| | **Total III** | **31** | **31** | **3** | **28** |
| LinkedIn | Government | 0 | 0 | 0 | 0 |
| | NGOs | 0 | 0 | 0 | 0 |
| | Private Firms | 15 | 7 | 1 | 6 |
| | Others | 11 | 4 | 0 | 4 |
| | **Total IV** | **26** | **11** | **1** | **10** |
| **Total** | **I+II+III+IV** | **174** | **118** | **16** | **102** |

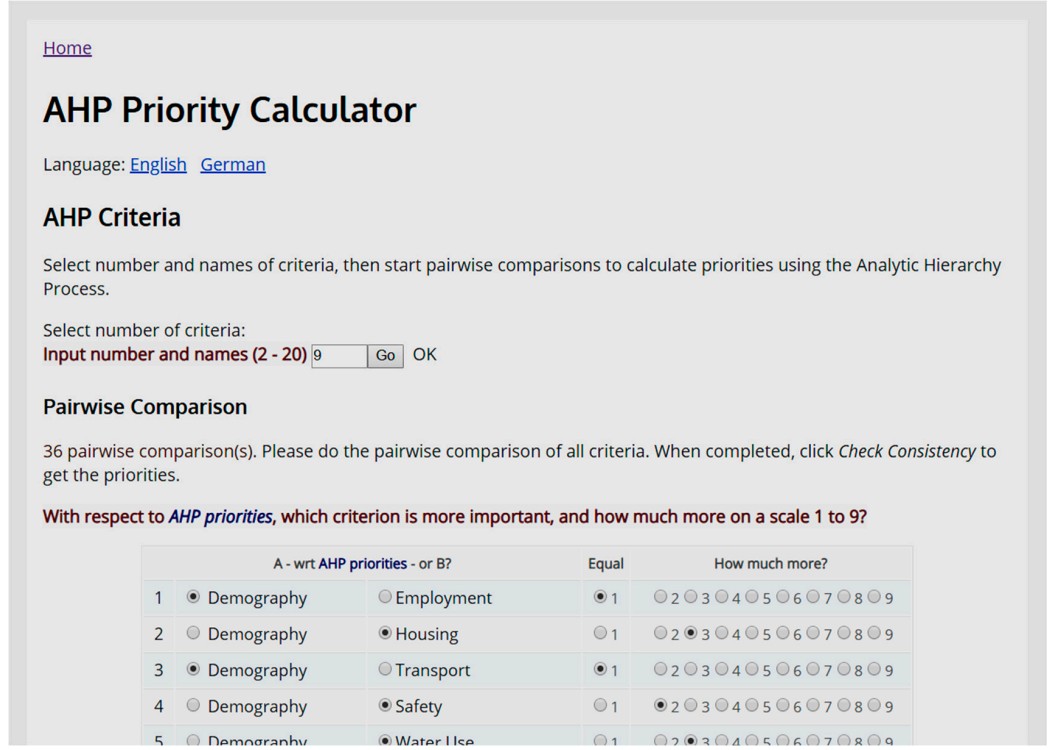

**Figure 2.** AHP-OS priority calculator.

The significant roles of the four-survey methods are as follows. Firstly, the Paper (face-to-face interview) method was used to survey the relevant government officials, especially senior officials. As shown in the table above, most of the valid samples from the governmental institutions were obtained from the paper survey. This method was also an alternative when observing that most of the distributed questionnaires through emails unlikely to reply (see Figure 3). Furthermore, this survey method also gave a chance to explain well about the AHP questionnaires to the respondents, and when the respondents were not sure about the questionnaires, they could ask immediately; consequently, this survey method obtained less inconsistent samples. Among the 39 gathered questionnaires, there are only two samples inconsistent (5.13%). However, this method was not really easy in terms of travelling to meet the targeted respondents and busy and limited time of them. Therefore, the online-survey methods are the best options to deal with these situations.

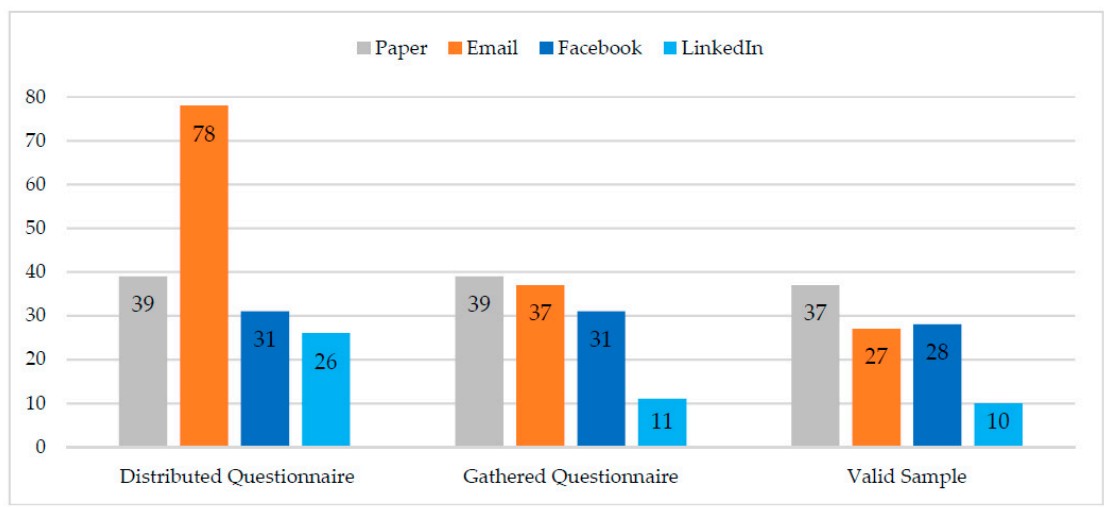

**Figure 3.** The number of distributed, gathered, and valid questionnaires by the survey methods.

According to the table above, the questionnaires distributed through emails are much more than other methods. Among the 174 (total) distributed questionnaires through all methods, there were 78 (44.83%) questionnaires distributed through emails. The email addresses were obtained from the Delphi surveys and the group emails of the green and sustainable city development program in Cambodia. The group emails contain the email addresses of the representatives from many organizations and agencies from all governmental and non-governmental organizations, private firms and companies, and others (research institutions, independent researchers, research students, experienced retirees, activists, and youth) experienced and working in the field of urban planning, development, management, and assessment in Cambodia. P.C. is in those groups as he has been one of the facilitators from the Ministry of Environment for the above green and sustainable city development program. There are more than 100 emails in the groups, but this research selected only the relevant ones for sending them questionnaires. Among 78 sent emails, we gathered 37 questionnaires and obtained 27 consistent samples. It means there were 10 gathered questionnaires inconsistent, equal to 27.03% (10/37). Another online-survey method is Facebook. Facebook is very popular and wisely used in Cambodia. It is also used in many purposes, including the surveys as it is very fast and easy to use and follow up. Furthermore, as P.C. has many friends as potential respondents on Facebook, this means was also an important means to obtain more potential respondents. There were 31 questionnaires sent to the well-known urban planners, Cambodia's city assessment specialists, independent urban researchers, urban research students, real estate officers, architects, and constructors. As they are well-known respondents, and Facebook is easy to follow up, all of the 31 distributed questionnaires were gathered. More importantly, this method allowed the respondents to ask the questions easily and quickly after receiving the questionnaire and having questions. There were a few

respondents who used the voice record function to ask the questions as well. Among 31 gathered questionnaires, there were three inconsistent, equal to 9.68% (3/31). The final online-survey method was LinkedIn. Even though LinkedIn is not popular for general people, it is popular for professional specialists, especially specialists in the fields of real estates. LinkedIn is also searchable for the fields and working addresses of the specialists. It generally shows us the levels of specialists' education, expertise, and positions. This method was used to survey the respondents from the relevant private firms, independent research institutions, and especially real estate companies. As it is not popularly used by the governmental and non-governmental organizations in Cambodia, this research did not use LinkedIn to survey the respondents from these organizations. As shown in the above figure, there were the 26 questionnaires sent to potential respondents through LinkedIn. We gathered 11 questionnaires from them. Among the 11 gathered questionnaires, there was only one questionnaire inconsistent, which is equal to 9.10% (1/11).

By using the above four-survey methods, this research gathered 118 questionnaires. Among these gathered questionnaires, there were 16 inconsistent (consistency ratio > 0.1), equal to 13.56%. Therefore, this research remained with 102 valid samples (consistent samples), equal to 86.44%. These valid samples classified by the types of institutions are shown in Figure 4. This figure shows that 42% of the sample size, equal to 43 respondents, were from the relevant governmental institutions. Furthermore, 21% of the sample size, equal to 22 respondents, were from the relevant non-governmental organizations. Moreover, 21% of the sample size, equal to 21 respondents, were from the relevant private firms and companies. Finally, 16% of the sample size, equal to 16 respondents, were other potential respondents such as independent researchers, research students, experienced retirees, and activists. Since the online survey started from 5 June 2019, offline survey was from 13 July 2019 to 11 August 2019, until obtaining the 102 valid samples on 2 September 2019, this research took three months to get the required sample size.

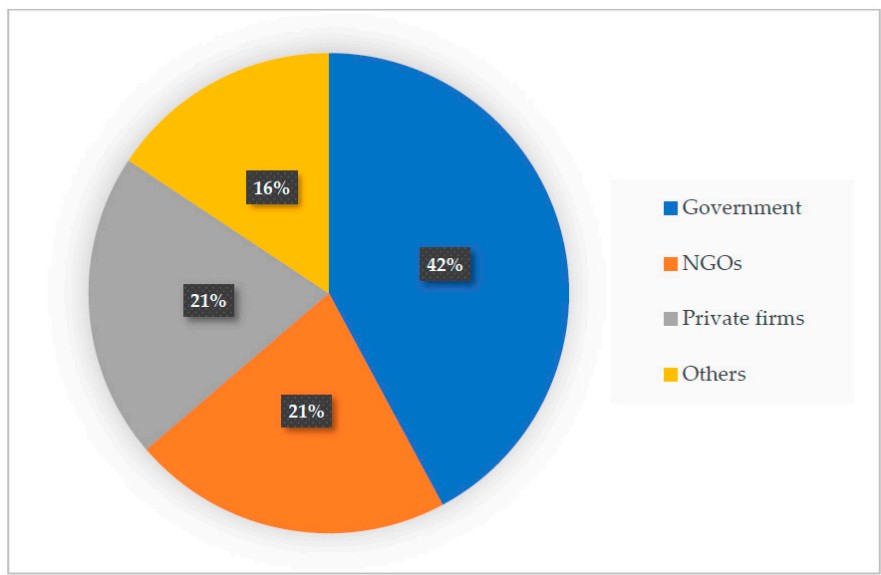

**Figure 4.** Percentage of the 102 valid samples classified by the type of working places.

## 3. Results

The results of this research illustrated in three parts according to the step-by-step calculations. The first part presented the priority weights of the nine criteria. The second part presented the priority weights of the assessment indicators in each criterion. The third part presented the priority weights of the 32 assessment indicators from all criteria. As mentioned in Section 2, the results of the first and second parts were calculated through the Microsoft Excel and AHP-OS online program. A consistent sample calculated by this program illustrated in Appendix C. More importantly, the results of the third

part were calculated by multiplying the priority weight of each indicator with the priority weight of their criteria. The priority weights in this part named total weights. All of the obtained priority weights from the first, second, and third parts, including the rank of the 32 assessment indicators by their total weights are shown in Table 3.

**Table 3.** Priority weights of the nine criteria and their assessment indicators and total weights of all indicator and their rank.

| Criteria | Weight | Indicator | Weight | Total Weight | Rank |
|---|---|---|---|---|---|
| Demography | 0.112 | Population density | 0.260 | 0.0291 | 17 |
| | | City's migration | 0.413 | 0.0462 | 5 |
| | | Household income | 0.327 | 0.0366 | 11 |
| Employment | 0.128 | Labor forces | 0.328 | 0.0421 | 8 |
| | | Unemployment | 0.403 | 0.0516 | 2 |
| | | New jobs creation | 0.269 | 0.0345 | 15 |
| Housing | 0.136 | Slum population | 0.410 | 0.0557 | 1 |
| | | Low-income housing | 0.328 | 0.0445 | 6 |
| | | Quality of buildings | 0.262 | 0.0356 | 13 |
| Transport | 0.111 | Transport means | 0.242 | 0.0268 | 21 |
| | | Sidewalks | 0.211 | 0.0234 | 23 |
| | | Parking lots | 0.189 | 0.0209 | 25 |
| | | Traffic congestion | 0.359 | 0.0398 | 10 |
| Safety | 0.137 | Crime prevention | 0.342 | 0.0470 | 3 |
| | | Construction safety | 0.291 | 0.0400 | 9 |
| | | Disaster prevention | 0.199 | 0.0274 | 20 |
| | | Insurances | 0.172 | 0.0237 | 22 |
| Water Use | 0.115 | Water supply | 0.407 | 0.0469 | 4 |
| | | Water consumption | 0.278 | 0.0320 | 16 |
| | | Water reservoirs | 0.315 | 0.0362 | 12 |
| Waste Management | 0.107 | Solid waste collection | 0.408 | 0.0437 | 7 |
| | | Wastewater treatment | 0.328 | 0.0352 | 14 |
| | | Waste reduction | 0.265 | 0.0284 | 19 |
| Air Quality and Energy | 0.074 | Fine dust levels | 0.168 | 0.0124 | 30 |
| | | Urban forests | 0.204 | 0.0150 | 29 |
| | | Energy consumption | 0.386 | 0.0284 | 18 |
| | | Renewable energy | 0.242 | 0.0178 | 27 |
| Urban Spaces and Tourism | 0.080 | Urban parks | 0.287 | 0.0229 | 24 |
| | | Botanic gardens | 0.108 | 0.0086 | 32 |
| | | Heritage conservation | 0.245 | 0.0195 | 26 |
| | | Tourism growth | 0.217 | 0.0173 | 28 |
| | | Playgrounds | 0.143 | 0.0114 | 31 |
| **Total** | **1.000** | **-** | **9.000** | **1.000** | **-** |

## 3.1. Priority Weights of the Nine Criteria

The priority weights of the nine criteria are as follows. "Demography" is 0.112, equal to 11.2% in percentage. "Employment" is 0.128, equal to 12.8% in percentage. "Housing" is 0.136, equal to 13.6% in percentage. "Transport" is 0.111, equal to 11.1% in percentage. "Safety" is 0.137, equal to 13.7% in percentage. "Water Use" is 0.115, equal to 11.5% in percentage. "Waste management" is 0.107, equal to 10.7% in percentage. "Air Quality and Energy" is 0.074, equal to 7.4% in percentage. "Urban Spaces and Tourism" is 0.080, equal to 8.0% in percentage. These relative weights are shown in Figure 5. Based on these weights, the nine assessment criteria for Cambodia's cities are ranked as follows. "Safety" is at the first rank. "Housing" is at the second rank. "Employment" is at the third rank. "Water Use" is at the fourth rank. "Demography" is at the fifth rank. "Transport" is at the sixth rank. "Waste management" is at the seventh rank. "Urban Spaces and Tourism" is at the eighth rank. "Air Quality and Energy" is at the ninth rank.

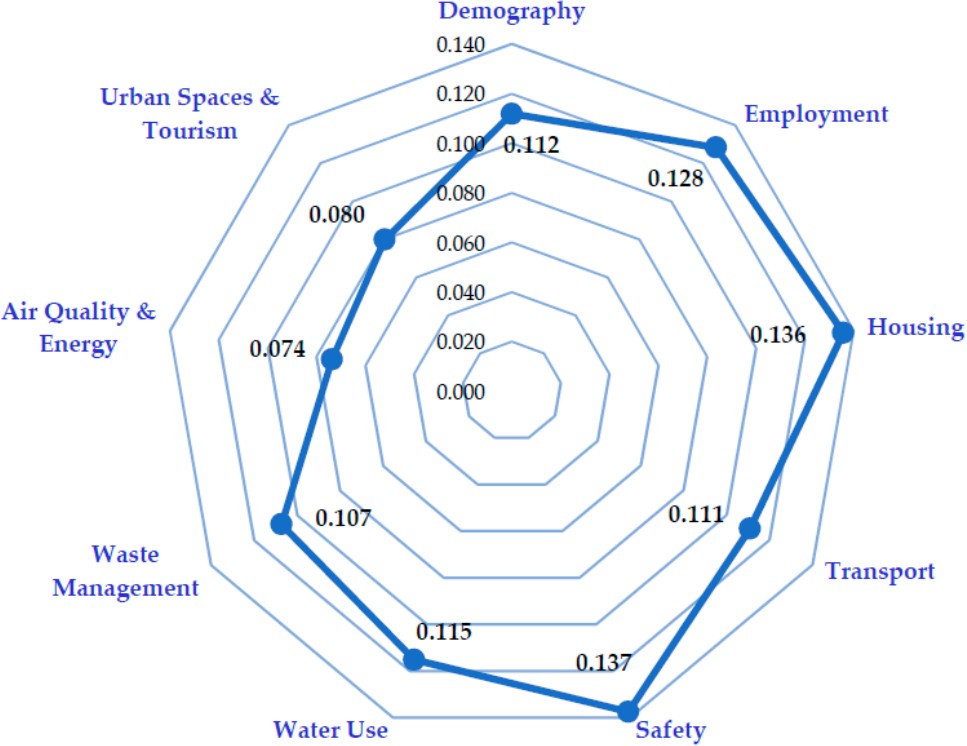

**Figure 5.** Relative weights of the nine criteria.

*3.2. Priority Weights of the Indicators in Each Criterion*

3.2.1. Demography

The priority weights of the assessment indicators for "Demography" are as follows. "Population density" is 0.260, equal to 26.0% in percentage. "City's migration" is 0.413, equal to 41.3% in percentage. "Household income" is 0.327, equal to 32.7% in percentage. These relative weights are shown in Figure 6. Based on these weights, the assessment indicators for "Demography" in Cambodia's cities are ranked as follows. "City's migration" is at the first rank. "Household income" is at the second rank. "Population density" is at the third rank.

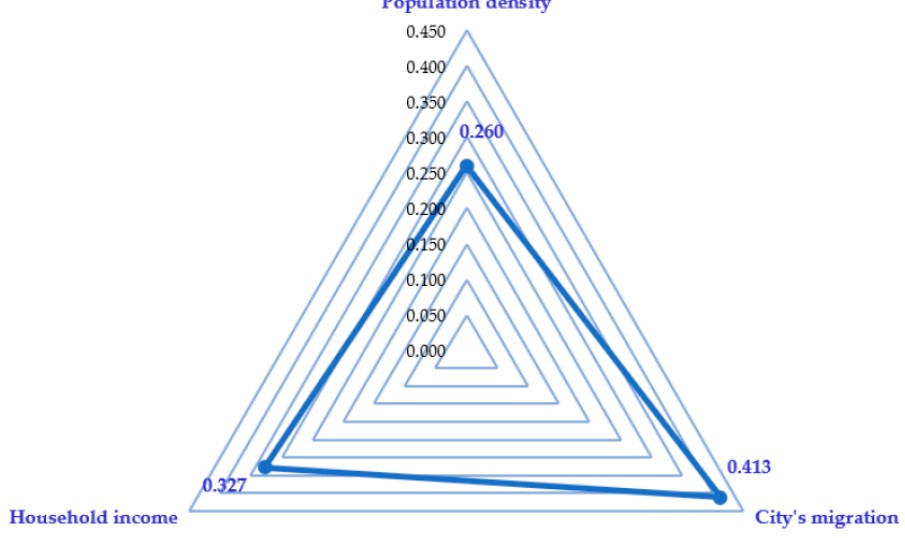

**Figure 6.** Relative weights of the assessment indicators for "Demography".

### 3.2.2. Employment

The priority weights of the assessment indicators for "Employment" are as follows. "Labor forces" is 0.328, equal to 32.8% in percentage. "Unemployment" is 0.403, equal to 40.3% in percentage. "New jobs creation" is 0.269, equal to 26.9% in percentage. These relative weights are shown in Figure 7. Based on these weights, the assessment indicators for "Employment" in Cambodia's cities are ranked as follows. "Unemployment" is at the first rank. "Labor forces" is at the second rank. "New jobs creation" is at the third rank.

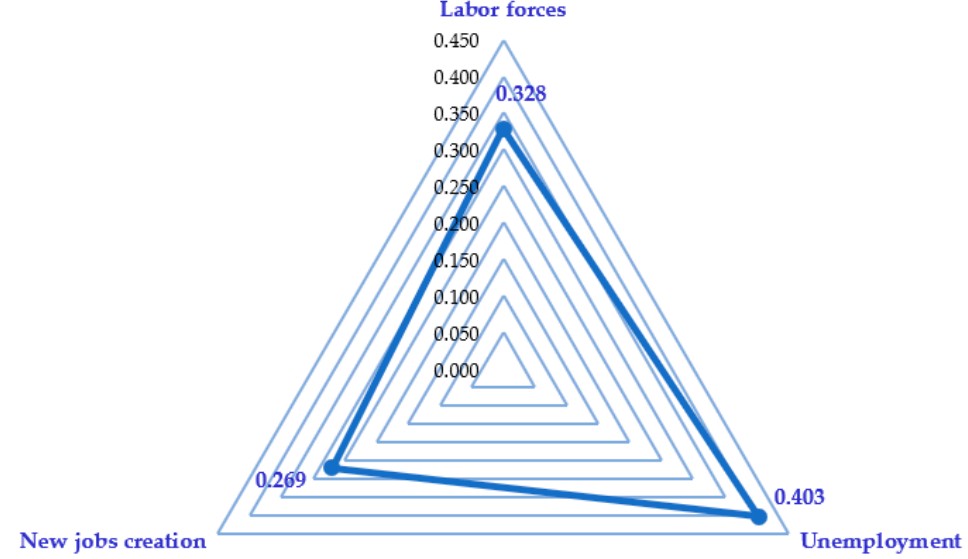

**Figure 7.** Relative weights of the assessment indicators for "Employment".

### 3.2.3. Housing

The priority weights of the assessment indicators for "Housing" are as follows. "Slum population" is 0.410, equal to 41.0% in percentage. "Low-income housing" is 0.328, equal to 32.8% in percentage. "Quality of buildings" is 0.262, equal to 26.2% in percentage. These relative weights are shown in Figure 8. Based on these weights, the assessment indicators for "Housing" in Cambodia's cities are ranked as follows. "Slum population" is at the first rank. "Low-income housing" is at the second rank. "Quality of buildings" is at the third rank.

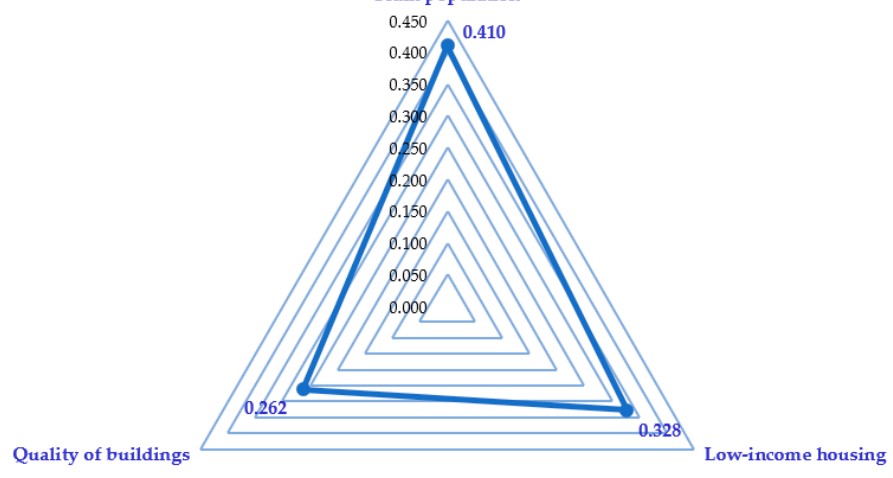

**Figure 8.** Relative weights of the assessment indicators for "Housing".

### 3.2.4. Transport

The priority weights of the assessment indicators for "Transport" are as follows. "Transport means" is 0.242, equal to 42.2% in percentage. "Sidewalks" is 0.211, equal to 21.1% in percentage. "Parking lots" is 0.189, equal to 18.9% in percentage. "Traffic congestion" is 0.359, equal to 35.9% in percentage. These relative weights are shown in Figure 9. Based on these weights, the assessment indicators for "Transport" in Cambodia's cities are ranked as follows. "Traffic congestion" is at the first rank. "Transport means" is at the second rank. "Sidewalks" is at the third rank. "Parking lots" is at the fourth rank.

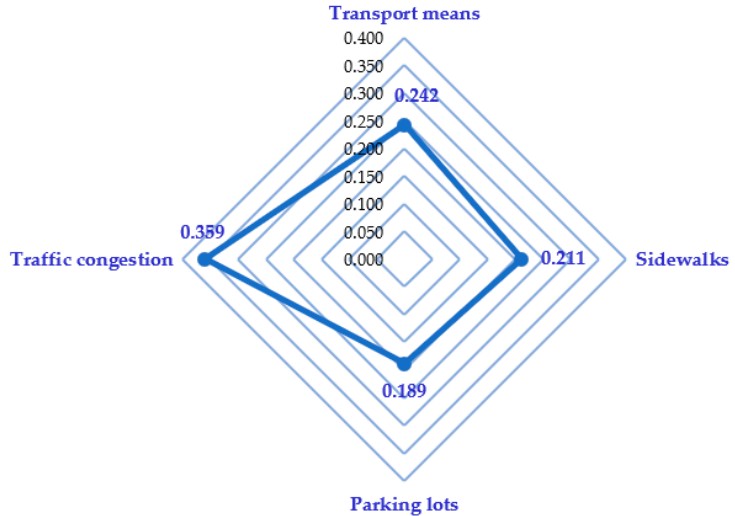

**Figure 9.** Relative weights of the assessment indicators for "Transport".

### 3.2.5. Safety

The priority weights of the assessment indicators for "Safety" are as follows. "Crime prevention" is 0.342, equal to 34.2% in percentage. "Construction safety" is 0.291, equal to 29.1% in percentage. "Disaster prevention" is 0.199, equal to 19.9% in percentage. "Insurances" is 0.172, equal to 17.2% in percentage. These relative weights are shown in Figure 10. Based on these weights, the assessment indicators for "Safety" in Cambodia's cities are ranked as follows. "Crime prevention" is at the first rank. "Construction safety" is at the second rank. "Disaster prevention" is at the third rank. "Insurances" is at the fourth rank.

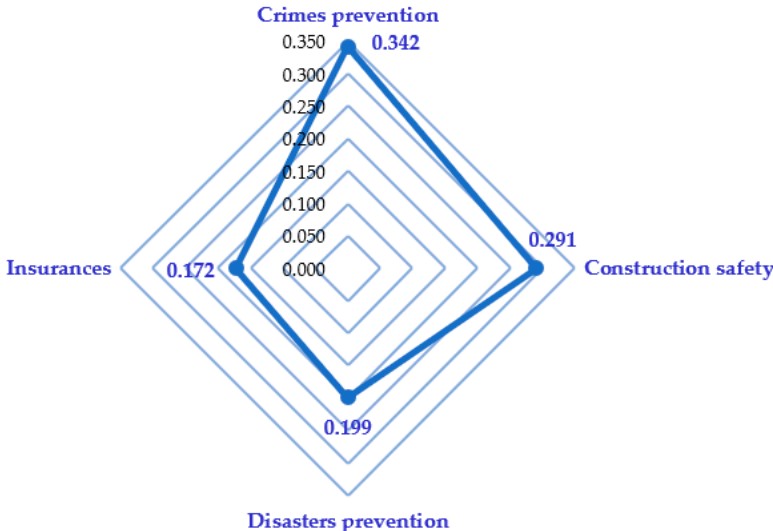

**Figure 10.** Relative weights of the assessment indicators for "Safety".

### 3.2.6. Water Use

The priority weights of the assessment indicators for "Water Use" are as follows. "Water supply" is 0.407, equal to 40.7% in percentage. "Water consumption" is 0.278, equal to 27.8% in percentage. "Water reservoirs" is 0.315, equal to 31.5% in percentage. These relative weights are shown in Figure 11. Based on these weights, the assessment indicators for "Water Use" in Cambodia's cities are ranked as follows. "Water supply" is at the first rank. "Water reservoirs" is at the second rank. "Water consumption" is at the third rank.

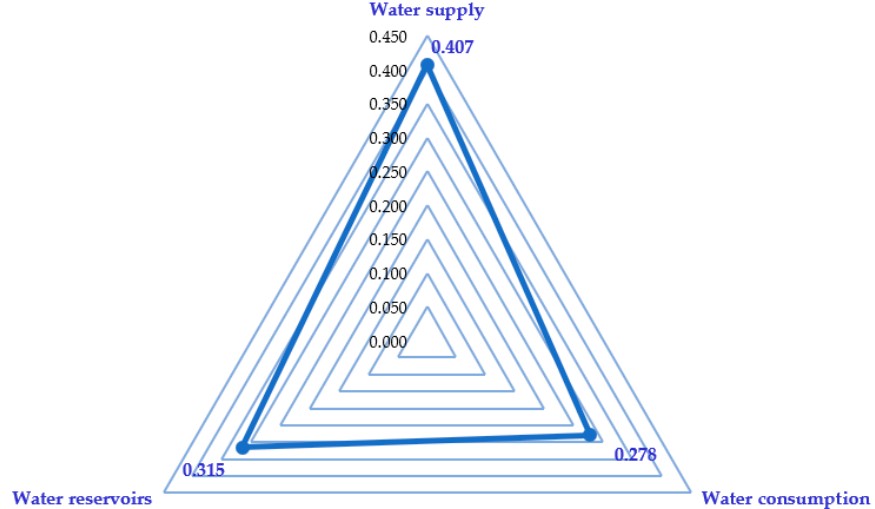

**Figure 11.** Relative weights of the assessment indicators for "Water Use".

### 3.2.7. Waste Management

The priority weights of the assessment indicators for "Waste management" are as follows. "Solid waste collection" is 0.408, equal to 40.8% in percentage. "Wastewater treatment" is 0.328, equal to 32.8% in percentage. "Waste reduction" is 0.265, equal to 26.5% in percentage. These relative weights are shown in Figure 12. Based on these weights, the assessment indicators for "Waste management" in Cambodia's cities are ranked as follows. "Solid waste collection" is at the first rank. "Wastewater treatment" is at the second rank. "Waste reduction" is at the third rank.

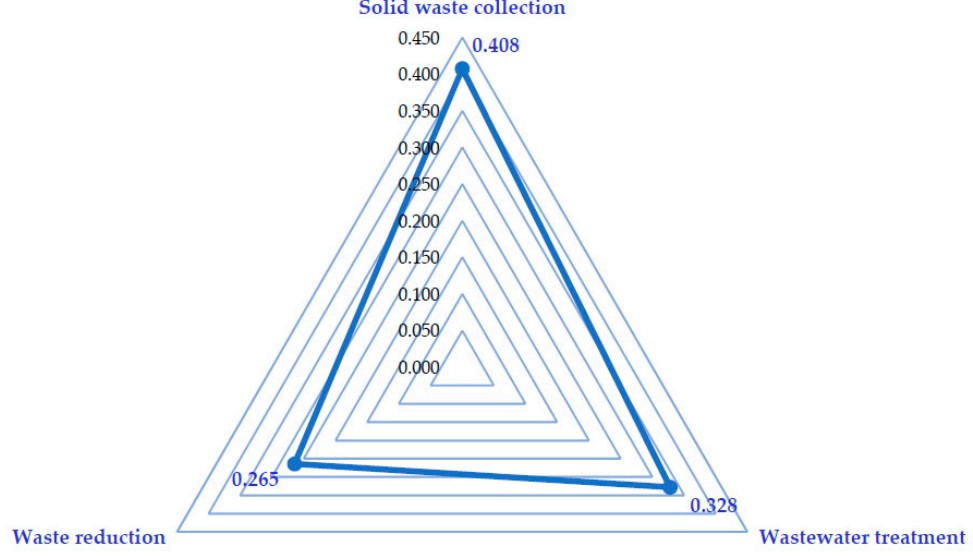

**Figure 12.** Relative weights of the assessment indicators for "Waste Management".

### 3.2.8. Air Quality and Energy

The priority weights of the assessment indicators for "Air Quality and Energy" are as follows. "Fine dust levels" is 0.168, equal to 16.8% in percentage. "Urban forests" is 0.204, equal to 20.4% in percentage. "Energy consumption" is 0.386, equal to 38.6% in percentage. "Renewable energy" is 0.242, equal to 24.2% in percentage. These relative weights are shown in Figure 13. Based on these weights, the assessment indicators for "Air Quality and Energy" in Cambodia's cities are ranked as follows. "Energy consumption" is at the first rank. "Renewable energy" is at the second rank. "Urban forests" is at the third rank. "Fine dust levels" is at the fourth rank.

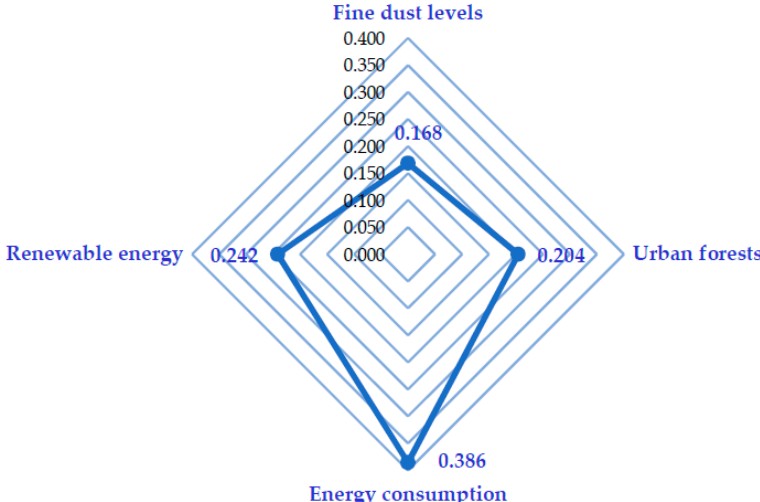

**Figure 13.** Relative weights of the assessment indicators for "Air Quality and Energy".

### 3.2.9. Urban Spaces and Tourism

The priority weights of the assessment indicators for "Urban Spaces and Tourism" are as follows. "Urban parks" is 0.287, equal to 28.7% in percentage. "Botanic gardens" is 0.108, equal to 10.8% in percentage. "Heritage conservation" is 0.245, equal to 24.5% in percentage. "Tourism growth" is 0.217, equal to 21.7% in percentage. "Playgrounds" is 0.143, equal to 14.3% in percentage. These relative weights are shown in Figure 14. Based on these weights, the assessment indicators for "Urban Spaces and Tourism" in Cambodia's cities are ranked as follows. "Urban parks" is at the first rank. "Heritage conservation" is at the second rank. "Tourism growth" is at the third rank. "Playgrounds" is at the fourth rank. "Botanic gardens" is at the fifth rank.

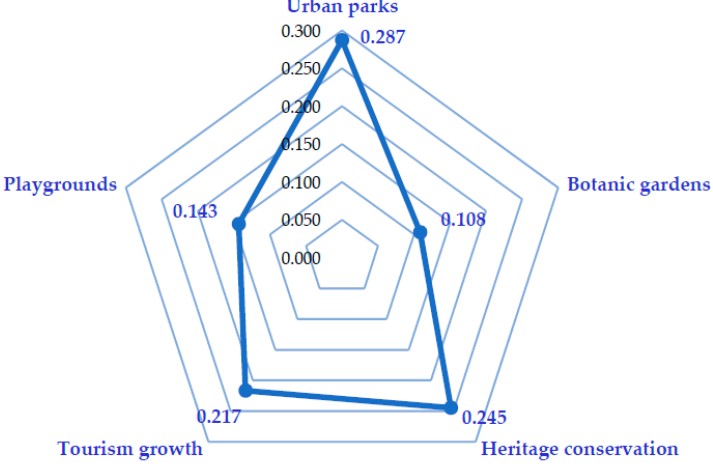

**Figure 14.** Relative weights of the assessment indicators for "Urban Spaces and Tourism".

### 3.3. Priority Weights of All 32 Indicators

The priority weights of the 32 assessment indicators relatively compared to each other are shown in Figure 15. The highest-priority indicator is "Slum population (0.0557 = 5.57%)" and the lowest-priority indicator is "Botanic gardens (0.0086 = 0.86%)". The order of all 32 assessment indicators from high to low priority is shown in Table 4. According to this table, there are two assessment indicators obtained priority weight above 0.05 (Rank no. 1 to 2). Furthermore, there are seven assessment indicators obtained priority weight in between 0.04 and 0.05 (Rank no. 3 to 9).

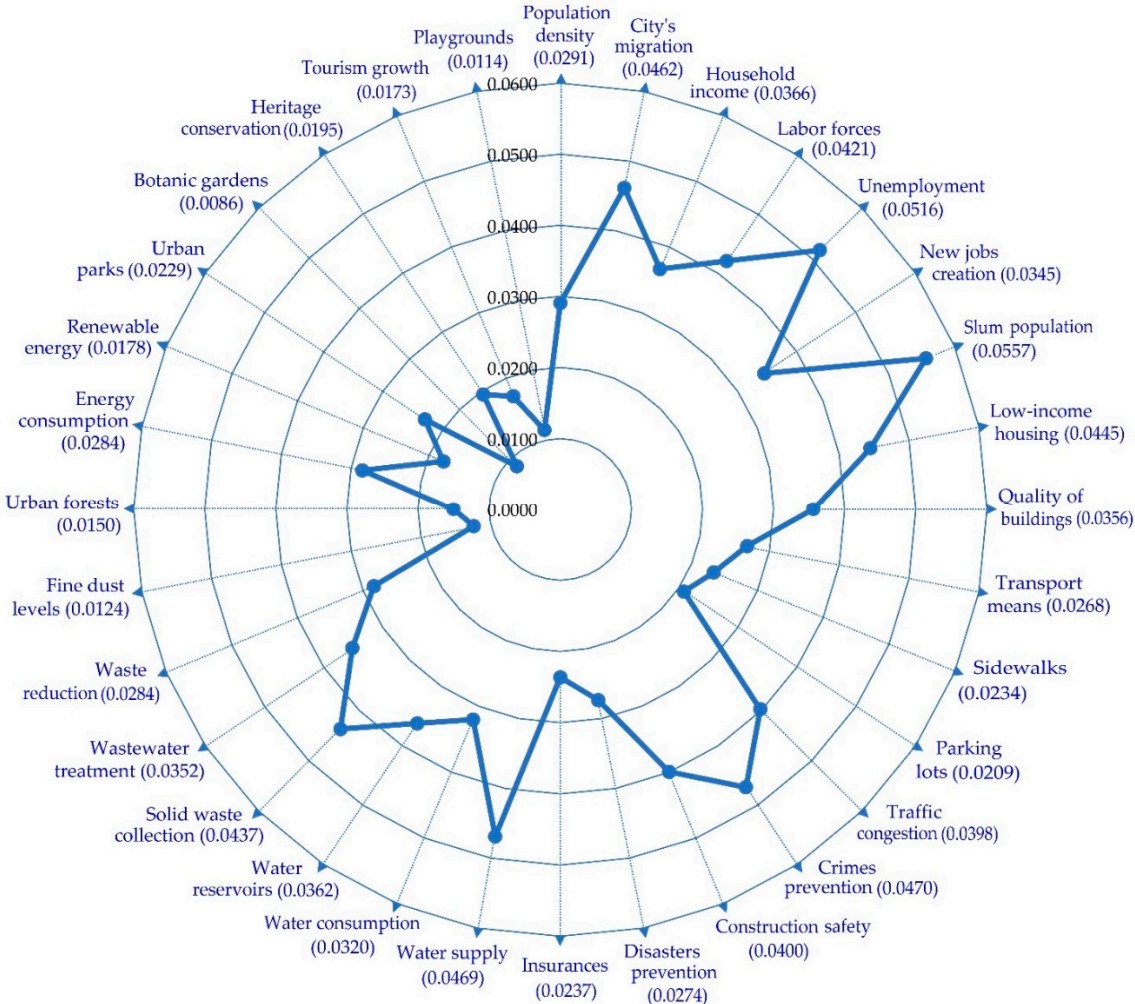

**Figure 15.** Relative weights of the 32 assessment indicators.

In addition, there are seven assessment indicators obtained priority weight between 0. 0.03 and 0.04 (Rank no. 10 to 16). Moreover, there are nine assessment indicators obtained priority weight between 0.02 and 0.03 (Rank no. 17 to 25). Furthermore, there are six assessment indicators obtained priority weight between 0.01 and 0.02 (Rank no. 26 to 31). Finally, there is one assessment indicator obtained priority weight below 0.01 (Rank no. 32). More importantly, as the average priority weight of assessment indicators is 0.0313 and according to Table 4, there are 16 assessment indicators (50%) above the average and 16 assessment indicators (50%) below the average.

**Table 4.** The order of the 32 assessment indicators from high to low priority.

| Indicator | Priority Weight | In Percentage | Rank |
|---|---|---|---|
| Slum population | 0.0557 | 5.57% | 1 |
| Unemployment | 0.0516 | 5.16% | 2 |
| Crime prevention | 0.0470 | 4.70% | 3 |
| Water supply | 0.0469 | 4.69% | 4 |
| City's migration | 0.0462 | 4.62% | 5 |
| Low-income housing | 0.0445 | 4.45% | 6 |
| Solid waste collection | 0.0437 | 4.37% | 7 |
| Labor forces | 0.0421 | 4.21% | 8 |
| Construction safety | 0.0400 | 4.00% | 9 |
| Traffic congestion | 0.0398 | 3.98% | 10 |
| Household income | 0.0366 | 3.66% | 11 |
| Water reservoirs | 0.0362 | 3.62% | 12 |
| Quality of buildings | 0.0356 | 3.56% | 13 |
| Wastewater treatment | 0.0352 | 3.52% | 14 |
| New jobs creation | 0.0345 | 3.45% | 15 |
| Water consumption | 0.0320 | 3.20% | 16 |
| Population density | 0.0291 | 2.91% | 17 |
| Energy consumption | 0.0284 | 2.84% | 18 |
| Waste reduction | 0.0284 | 2.84% | 19 |
| Disaster prevention | 0.0274 | 2.74% | 20 |
| Transport means | 0.0268 | 2.68% | 21 |
| Insurances | 0.0237 | 2.37% | 22 |
| Sidewalks | 0.0234 | 2.34% | 23 |
| Urban parks | 0.0229 | 2.29% | 24 |
| Parking lots | 0.0209 | 2.09% | 25 |
| Heritage conservation | 0.0195 | 1.95% | 26 |
| Renewable energy | 0.0178 | 1.78% | 27 |
| Tourism growth | 0.0173 | 1.73% | 28 |
| Urban forests | 0.0150 | 1.50% | 29 |
| Fine dust levels | 0.0124 | 1.24% | 30 |
| Playgrounds | 0.0114 | 1.14% | 31 |
| Botanic gardens | 0.0086 | 0.86% | 32 |
| **Total** | **1.000** | **100%** | **-** |

The top ten assessment indicators are as follows. (1) Slum population is 0.0557. (2) Unemployment is 0.0516. (3) Crime prevention is 0.0470. (4) Water supply is 0.0469. (5) City's migration is 0.0462. (6) Low-income housing is 0.0445. (7) Solid waste collection is 0.0437. (8) Labor-force is 0.0421. (9) Construction safety is 0.0400. (10) Traffic congestion is 0.0398. These assessment indicators and their description are shown in Table 5.

**Table 5.** Top 10 assessment indicators for sustainable city development and management in Cambodia.

| Rank | Indicator | Description |
|------|-----------|-------------|
| 1 | Slum population | This indicator assesses the residential and living environment in the city. It measures through a percentage of the population living in slums or informal/unplanned settlements. |
| 2 | Unemployment | This indicator assesses the employment situation in the city. It measures through the unemployment rate. |
| 3 | Crime prevention | This indicator assesses the city government's efforts in preventing crimes in the city. It measures through the number and type of measures or initiatives of the city government to prevent crimes. |
| 4 | Water supply | This indicator assesses the situation of water supply in the city. It measures through a percentage of households with access to potable water supply infrastructure compared to total households. |
| 5 | City's migration | This indicator assesses the migration situation in the city. It measures through the rural-urban migration rate in a year. |
| 6 | Low-income housing | This indicator assesses the city government's efforts in providing affordable housing. It measures through the number, type, and size of the low-income housing development projects. |
| 7 | Solid waste collection | This indicator assesses the public organizations for solid waste collection and the city government's efforts in improvement. It measures through a percentage of households linked to collecting network and number and type of improvement initiatives. |
| 8 | Labor forces | This indicator assesses the productivity of the city. It measures through the productive population in the labor forces. |
| 9 | Construction safety | This indicator assesses the city government's efforts in preventing construction risks. It measures through the number and type of initiated programs or activities to prevent construction risks. |
| 10 | Traffic congestion | This indicator assesses the city government's efforts in reducing traffic congestion. It measures through the number, type, and size of initiated programs or activities for reducing traffic congestion. |

## 4. Discussion

The results showed that the first priority criteria for sustainable city assessment in Cambodia is "Safety". Its priority weight is 0.137. The priority weights of its assessment indicators ranked from high to low are 0.342 for crime prevention, 0.291 for construction safety, 0.199 for disaster prevention, and 0.172 for social welfare registration. These priority weights are important for assessing and ranking sustainable urban safety of Cambodian cities. These priority weights are also significant to policy making, strategic direction, and budget allocation for sustainable urban safety system development and management in Cambodia. The second priority criteria for sustainable city assessment in Cambodia is "Housing". Its priority weight is 0.136. The priority weights of its assessment indicators ranked from high to low are 0.410 for residential and living environment improvement, 0.328 for low-income housing development, and 0.262 for quality residential buildings. These priority weights are important for assessing and ranking sustainable urban housing of Cambodian cities. These priority weights are also significant to policy making, strategic direction, and budget allocation for sustainable urban housing development and management in Cambodia. The third priority criteria for sustainable city assessment in Cambodia is "Employment". Its priority weight is 0.128. The priority weights of its assessment indicators ranked from high to low are 0.403 for unemployment reduction, 0.328 for productive labor forces, and 0.262 for new jobs creation. These priority weights are important for assessing and ranking sustainable urban employment of Cambodia cities. These priority weights are also significant to policy making, strategic direction, and budget allocation for sustainable urban employment development and management in Cambodia.

Furthermore, the fourth priority criteria for sustainable city assessment in Cambodia is "Water Use". Its priority weight is 0.115. The priority weights of its assessment indicators ranked from high to low are 0.407 for potable water supply infrastructure, 0.315 for freshwater supply sources, and 0.278

for water consumption. These priority weights are important for assessing and ranking sustainable urban water use of Cambodian cities. These priority weights are also significant to policy making, strategic direction, and budget allocation for sustainable urban water consumption and production in Cambodia. The fifth priority criteria for sustainable city assessment in Cambodia is "Demography". Its priority weight is 0.112. The priority weights of its assessment indicators ranked from high to low are 0.413 for rural-urban migration management, 0.327 for household income improvement, and 0.260 for living space improvement. These priority weights are important for assessing and ranking sustainable urban demographic structure of Cambodian cities. These priority weights are also significant to policy making, strategic direction, and budget allocation for sustainable urban demographic development and management in Cambodia. The sixth priority criteria for sustainable city assessment in Cambodia is "Transport". Its priority weight is 0.111. The priority weights of its assessment indicators ranked from high to low are 0.359 for traffic congestion reduction, 0.242 for public transport sharing rate, 0.211 for sidewalks improvement, and 0.189 for public parking lots preparation, management, and improvement. These priority weights are important for assessing and ranking sustainable urban transport of Cambodian cities. These priority weights are also significant to policy making, strategic direction, and budget allocation for sustainable urban transport development and management in Cambodia.

Moreover, the seventh priority criteria for sustainable city assessment in Cambodia is "Waste Management". Its priority weight is 0.107. The priority weights of its assessment indicators ranked from high to low are 0.408 for solid waste collection, 0.328 for wastewater treatment, and 0.265 for waste reduction. These priority weights are important for assessing and ranking sustainable urban waste management of Cambodian cities. These priority weights are also significant to policy making, strategic direction, and budget allocation for sustainable urban waste management in Cambodia. The eighth priority criteria for sustainable city assessment in Cambodia is "Urban Spaces and Tourism". Its priority weight is 0.080. The priority weights of its assessment indicators ranked from high to low are 0.287 for public and green spaces, 0.245 for cultural-historical and heritage conservation, 0.217 for tourism attraction and satisfaction, 0.143 for playground and leisure development, and 0.108 for biodiversity conservative parks/gardens preparation. These priority weights are important for assessing and ranking sustainable urban spaces and tourism of Cambodian cities. These weights are also significant to policy making, strategic direction, and budget allocation for sustainable urban spaces and tourism development and management in Cambodia. The ninth priority criteria for sustainable city assessment in Cambodia is "Air Quality and Energy". Its priority weight is 0.074. The priority weights of its assessment indicators ranked from high to low are 0.386 for energy-efficient use and saving, 0.242 for renewable energy use and promotion, 0.204 for urban forest conservation and plantation, and 0.168 for air quality improvement. These priority weights are important for assessing and ranking sustainable urban air quality and energy of Cambodian cities. These priority weights are also significant to policy making, strategic direction, and budget allocation for air quality improvement and sustainable urban energy consumption and production in Cambodia.

More importantly, this research revealed the total weights of all 32 assessment indicators for the development and management of sustainable cities in Cambodia. These total weights are important for inclusive assessing and ranking the urban sustainability of Cambodia's cities. These weights are also significant to policy making, strategic direction, and budget allocation for inclusive development and management of sustainable cities in Cambodia. Especially, the top ten indicators shown in Table 5 must be considered for monitoring and assessment in order to strongly improve urban sustainability as well as the ranking of sustainable cities in Cambodia. Furthermore, the previous research [1] showed that UN Sustainable Development Goal 11 (SDG 11) has nine indicators suitable for applying to Cambodia. Therefore, this research consequently prioritized and ranked these indicators based on the total weights of the correlated consensus indicators as shown in Table 6.

**Table 6.** Nine UN SDG 11 indicators prioritized for applying to Cambodia based on the priority weights of the correlated consensus indicators.

| Rank | Reviewed Indicators | Weight |
|:---:|:---|:---:|
| 1 | Percentage of the population living in slums or informal settlements | 0.0557 |
| 2 | Unemployment rate | 0.0516 |
| 3 | Percentage of households with access to potable water infrastructure | 0.0469 |
| 4 | Percentage of solid waste regularly collected | 0.0437 |
| 5 | Population density | 0.0291 |
| 6 | Disaster prevention | 0.0274 |
| 7 | Public transport sharing rate | 0.0268 |
| 8 | Budget provided to heritage conservation | 0.0195 |
| 9 | Fine dust levels (PM 2.5 or PM 10) | 0.0124 |

Furthermore, the mean values (levels of importance) of the 32 consensus indicators verified by Delphi and their rank based on these mean values are shown in Table 7. According to this table, the rank of the 32 indicators based on mean values are completely different from total weight's rank. Therefore, this research confirms that the levels of importance verified by Delphi cannot be used for prioritizing or ranking the consensus indicators. As explained in Delphi's characteristic [1–3], the level of importance shows how each indicator is important individually, not how each indicator is important comparatively with all indicators like the relative weight verified by AHP.

**Table 7.** AHP's and Delphi's ranks of the 32 assessment indicators.

| Indicator | AHP | | Delphi | |
|:---|:---:|:---:|:---:|:---:|
| | Weight | Rank | Mean | Rank |
| Slum population | 0.0557 | 1 | 3.6 | 28 |
| Unemployment | 0.0516 | 2 | 3.5 | 31 |
| Crime prevention | 0.0470 | 3 | 4.2 | 10 |
| Water supply | 0.0469 | 4 | 4.6 | 3 |
| City's migration | 0.0462 | 5 | 3.6 | 28 |
| Low-income housing | 0.0445 | 6 | 4.7 | 2 |
| Solid waste collection | 0.0437 | 7 | 4.6 | 3 |
| Labor forces | 0.0421 | 8 | 4.2 | 10 |
| Construction safety | 0.0400 | 9 | 3.9 | 18 |
| Traffic congestion | 0.0398 | 10 | 3.8 | 24 |
| Household income | 0.0366 | 11 | 4.2 | 10 |
| Water reservoirs | 0.0362 | 12 | 3.9 | 18 |
| Quality of buildings | 0.0356 | 13 | 3.9 | 18 |
| Wastewater treatment | 0.0352 | 14 | 4.1 | 13 |
| New jobs creation | 0.0345 | 15 | 4.6 | 3 |
| Water consumption | 0.0320 | 16 | 4.5 | 7 |
| Population density | 0.0291 | 17 | 3.9 | 18 |
| Energy consumption | 0.0284 | 18 | 3.9 | 18 |
| Waste reduction | 0.0284 | 19 | 4.3 | 9 |
| Disaster prevention | 0.0274 | 20 | 3.7 | 25 |
| Transport means | 0.0268 | 21 | 3.9 | 18 |
| Insurances | 0.0237 | 22 | 3.5 | 31 |
| Sidewalks | 0.0234 | 23 | 4.8 | 1 |
| Urban parks | 0.0229 | 24 | 4.1 | 13 |
| Parking lots | 0.0209 | 25 | 4.1 | 13 |
| Heritage conservation | 0.0195 | 26 | 3.7 | 25 |
| Renewable energy | 0.0178 | 27 | 3.7 | 25 |
| Tourism growth | 0.0173 | 28 | 4.0 | 17 |
| Urban forests | 0.0150 | 29 | 4.5 | 7 |
| Fine dust levels | 0.0124 | 30 | 4.6 | 3 |
| Playgrounds | 0.0114 | 31 | 3.6 | 28 |
| Botanic gardens | 0.0086 | 32 | 4.1 | 13 |

## 5. Summary and Conclusions

Through the prioritization of the 32 assessment indicators by firstly prioritizing the nine criteria and then their assessment indicators, this research obtained the priority weights as follows. Firstly, the priority weights of the nine criteria ranked from high to low are Safety (0.137), Housing (0.136), Employment (0.128), Water Use (0.115), Demography (0.112), Transport (0.111), Waste management (0.107), Urban Spaces and Tourism (0.080), Air Quality and Energy (0.074). Furthermore, the priority weights of the assessment indicators in each criterion ranked from high to low are as follows. "Safety" obtained 0.342 for crime prevention, 0.291 for construction safety, 0.199 for disaster prevention, and 0.172 for social welfare registration. "Housing" obtained 0.410 for residential and living environment improvement, 0.328 for low-income housing development, and 0.262 for quality residential buildings. "Employment" obtained 0.403 for unemployment reduction, 0.328 for productive labor forces, and 0.262 for new jobs creation. "Water Use" obtained 0.407 for potable water supply infrastructure, 0.315 for freshwater supply sources, and 0.278 for water consumption. "Demography" obtained 0.413 for rural-urban migration management, 0.327 for household income improvement, and 0.260 for living spaces improvement. "Transport" obtained 0.359 for traffic congestion reduction, 0.242 for public transport sharing rate, 0.211 for sidewalks improvement, and 0.189 for public parking lots management and improvement. "Waste Management" obtained 0.408 for solid waste collection, 0.328 for wastewater treatment, and 0.265 for waste reduction. "Urban Spaces and Tourism" obtained 0.287 for public and green spaces, 0.245 for cultural-historical and heritage conservation, 0.217 for tourism attraction and satisfaction, 0.143 for playground and leisure development, and 0.108 for biodiversity conservative parks/gardens preparation. "Air Quality and Energy" obtained 0.386 for energy-efficient use and saving, 0.242 for renewable energy use and promotion, 0.204 for urban forest conservation and plantation, and 0.168 for air quality improvement. These priority weights are important for ranking sectoral development and management of sustainable cities in Cambodia. Likewise, these priority weights are also significant to policy making, strategic direction, and budget allocation for sectoral development and management of sustainable cities in Cambodia.

Importantly, the 32 assessment indicators obtained total prioritized weights as follows. There are two indicators that obtained priority weight above 0.05. Furthermore, there are seven indicators obtained priority weight between 0.04 and 0.05. In addition, there are seven indicators obtained priority weight between 0. 0.03 and 0.04. Moreover, there are nine indicators obtained priority weight between 0.02 and 0.03. Likewise, there are six indicators obtained priority weight between 0.01 and 0.02. Finally, there is one indicator that obtained priority weight below 0.01. Significantly, the top ten indicators ranked based on the total prioritized weights are 'Residential and living environment management' (0.0557), 'Unemployment reduction' (0.0516), 'Crime prevention' (0.0470), 'Potable water supply infrastructure' (0.0469), 'Rural-urban migration management' (0.0462), 'Low-income housing development' (0.0445), 'Solid waste collection' (0.0437), 'Productive labor forces' (0.0421), 'Construction safety' (0.0400), and 'Traffic congestion reduction' (0.0398). The average priority weight of all indicators is 0.0313, and half of them obtained priority weight above the average.

The total weights of the 32 assessment indicators are important for the development and management of sustainable cities in Cambodia. Primarily, these weights are important for (a) inclusive monitoring and assessment of the development and management of sustainable cities in Cambodia and (b) ranking the urban sustainability of Cambodian cities. Furthermore, these prioritized weights are significant to policy making, strategic direction, and budget allocation for inclusive development and management of sustainable cities. This research suggests using these indicators for monitoring and assessment of cities in Cambodia. Especially, the top ten indicators must be considered to use for monitoring and assessment in order to strongly improve urban sustainability. In the case of sustainable city contests, achieving these top indicators, the city can obtain good scores and rank. In addition, this research recommends selecting and applying the UN SDG 11's indicators to Cambodia based on (i) the correlation of its indicators with the consensus indicators and (ii) the weights of the consensus indicators that correlated with its indicators.

The rank of the 32 consensus indicators based on the levels of importance verified by the Delphi technique in previous research are completely different from the total weight's rank. Therefore, this research confirms that the levels of importance verified by the Delphi technique cannot be used for prioritizing or ranking the consensus indicators because the level of importance defines how each indicator is important individually based on the experienced panelists' opinions, not how each indicator is important comparatively with all indicators like the total weights verified by the AHP technique. In this prioritization research, the following could be the limitations. As the AHP technique is very new to Cambodia, especially for the respondents there were limitations in terms of understanding—besides offline survey, online survey methods were challenging in that inconsistent samples were obtained. Therefore, in order to improve accuracy, face-to-face survey methods using panel groups or workshops would be the best option because in these methods the AHP technique and its questionnaires can be well explained to the respondents and their feedback can be received immediately. In this case, this research recommends that future AHP research in Cambodia must be considered on the survey methods. For the government projects, however, using the AHP technique will not be challenging because the government usually hosts the meeting and/or workshop with all relevant stakeholders for their research and development projects. Therefore, the survey through the meeting or workshop will be more consistent and accurate.

**Author Contributions:** Conceptualization, P.C.; data curation, P.C.; formal analysis, P.C.; funding acquisition, P.C.; investigation, P.C.; methodology, P.C.; project administration, P.C. and M.-H.L.; resources, P.C. and M.-H.L.; software, P.C.; supervision, M.-H.L.; validation, P.C. and M.-H.L.; visualization, P.C. and M.-H.L.; writing—original draft, P.C.; writing—review and editing, P.C. and M.-H.L.

**Funding:** This research received no external funding.

**Acknowledgments:** The authors would like to thank the Cambodian Ministry of Environment, Department of Green Economy, and General Secretariat of the National Council for Sustainable Development for administratively supporting this research, and National Institute of International Education of the Korean Ministry of Education for providing doctoral research scholarships to P.C. which were allocated for this research.

**Conflicts of Interest:** The author declares no conflict of interest.

## Appendix A. Relevant Indicators

**Table A1.** Review and classification of the five major source indicators.

| Category | Indicator | SDG11 | ESC | HAN | GC | CC |
|---|---|---|---|---|---|---|
| Demography | Population density | ● | | ● | | |
| | Population growth rate | ● | | ● | | |
| | The ratio of land consumption rate to the population growth rate | ● | | ● | | |
| | Birth rate | | | ● | | |
| | Active population (20–65) rate | | | ● | | |
| | Elderly population (over 65) rate | | | ● | | |
| Jobs and Tourism | Labor force participation rate | | | ● | | |
| | Unemployment rate | ● | | ● | | |
| | Absence or presence of shopping centers or shopping outlets within tourist areas | | | | | ● |
| | Absence or presence of local products in shopping centers within tourist areas | | | | | ● |
| | Tourism growth rate per year | | | | ● | |
| | Number of registered foreigners | | | ● | | |
| Housing | Percentage of the population living in slums | ● | | ● | | |
| | Percentage of the population spending more than 30% of their income on housing costs | | | ● | | |
| | Number of low-income housing units | | | | ● | |
| | Percentage of population living in owned houses | | | ● | | |
| | Percentage of aging residential buildings | | | ● | | |

**Table A1.** *Cont.*

| Category | Indicator | SDG11 | ESC | HAN | GC | CC |
|---|---|:---:|:---:|:---:|:---:|:---:|
| Transport | Percentage of population living within 0.5 km of public transport access | ● | | | | |
| | Percentage of people using large public transports | ● | | | | |
| | Public transport sharing rate | | | ● | | |
| | Investment in transports under the budget | ● | | ● | | |
| | Number of taxi and bus | | | ● | | |
| | Proportion of environmentally friendly vehicles | | | | | ● |
| | Proportion of traffic congestion level (extra hours of travel time) | | | | ● | ● |
| Safety | Absence or presence of measures to prevent crimes | | | ● | | ● |
| | Number of people affected by crimes; number of crimes | | | ● | | ● |
| | Number of affected people resulting from disasters | ● | | | | |
| | Number of damaged or destroyed houses | ● | | ● | | |
| | Proportion of disaster prevention facilities (constructed dams, reservoirs, pumping stations etc.) | ● | | ● | ● | |
| | Proportion of basic livelihood security | | | ● | | |
| Clean Air and Energy | Fine dust level | ● | ● | ● | | |
| | Number of days in a year that Pollutant Standards Index (PSI) exceeded 100 (unhealthy) using USEPA standard | | ● | | | |
| | Proportion of the city government's efforts in greenhouse gas reduction | | | | ● | ● |
| | The ratio of forest conservation areas to the total land area of the city | | | ● | | |
| | Percentage of gasoline- and diesel-fueled vehicles that meet city or national standards during roadside inspection | | ● | | | |
| | Percentage of industries that fulfill the requirement of national standards | | ● | | ● | |
| | Types of alternative fuels used | | ● | | | |
| | Percentage of hotels using energy saving devices or renewable energy | | | | | ● |
| | Proportion of awareness campaigns on energy-saving techniques | | | | | ● |
| | Proportion of incentives for sustainable use of energy | | | | | ● |
| | Percentage of solar energy share in electricity supply | | | | ● | |
| | Number of buildings with solar PV installed | | | | ● | |
| | Number of green buildings in the city | ● | | | ● | |
| Waste Management | Percentage of solid waste regularly collected and recycled | ● | ● | ● | ● | ● |
| | Percentage of reduction in total waste generated a year | | ● | | | |
| | Percentage of waste collected from door to door/collection point | | ● | | | |
| | Percentage of waste transported in covered vehicles on a daily basis | | ● | | | |
| | Percentage of households and industries linked to sewerage system | | ● | | | |
| | Percentage of households with secured sanitation systems | | | | ● | |
| | Proportion of wastewater treatment plants in the city | | | | ● | |
| Water Use | Percentage of households with tap water that meets WHO drinking water standard | | ● | | | |
| | Percentage of school at all levels with water conservation education programs | | ● | | | |
| | Percentage of capacity of city in supplying water to meet average consumption | | ● | | | |
| | Percentage of available freshwater from ground and surface water extracted for use | | ● | | | |
| | Percentage of households with access to potable water infrastructure | | ● | ● | | |
| Public Space and Heritage | The ratio of public and green spaces compared to the total area of the city | ● | ● | ● | | |
| | Frequency and time of maintaining and cleaning public and green spaces | | | | | ● |
| | Percentage of the area that complies with the stipulated spatial plan of the city | | ● | ● | | |
| | Number of urban parks | | | ● | | |
| | Proportion of leisure areas in the city | | | | | ● |
| | Percentage of residents residing in public and green spaces accessible within 0.5 km | ● | | | | |
| | Percentage of conservation status given to historical and cultural areas | ● | | ● | | |
| | Percentage of the budget provided to maintain the cultural and natural heritage | ● | | ● | | |

**Note:** The point "●" shows where the indicators were reviewed from, SDG 11, ESC, HAN, GC, or CC. Furthermore, SDG 11 refers to UN sustainable development goal 11 indicators. ESC refers to ASEAN environmentally sustainable city indicators. HAN refers to Korean HAN case study indicators. GC refers to Cambodian green city indicators. CC refers to Cambodian clean city indicators.

## Appendix B. AHP Questionnaires

**Question 1:** Between the Criteria **A** and **B**, which one is more important for the assessment of sustainable city development and management in Cambodia?

| A | 9 | 8 | 7 | 6 | 5 | 4 | 3 | 2 | 1 | 2 | 3 | 4 | 5 | 6 | 7 | 8 | 9 | B |
|---|---|---|---|---|---|---|---|---|---|---|---|---|---|---|---|---|---|---|
| Demography | | | | | | | | | | | | | | | | | | Employment |
| Demography | | | | | | | | | | | | | | | | | | Housing |
| Demography | | | | | | | | | | | | | | | | | | Transport |
| Demography | | | | | | | | | | | | | | | | | | Safety |
| Demography | | | | | | | | | | | | | | | | | | Water Use |
| Demography | | | | | | | | | | | | | | | | | | Waste Management |
| Demography | | | | | | | | | | | | | | | | | | Air Quality & Energy |
| Demography | | | | | | | | | | | | | | | | | | Urban Spaces & Tourism |
| Employment | | | | | | | | | | | | | | | | | | Housing |
| Employment | | | | | | | | | | | | | | | | | | Transport |
| Employment | | | | | | | | | | | | | | | | | | Safety |
| Employment | | | | | | | | | | | | | | | | | | Water Use |
| Employment | | | | | | | | | | | | | | | | | | Waste Management |
| Employment | | | | | | | | | | | | | | | | | | Air Quality & Energy |
| Employment | | | | | | | | | | | | | | | | | | Urban Spaces & Tourism |
| Housing | | | | | | | | | | | | | | | | | | Transport |
| Housing | | | | | | | | | | | | | | | | | | Safety |
| Housing | | | | | | | | | | | | | | | | | | Water Use |
| Housing | | | | | | | | | | | | | | | | | | Waste Management |
| Housing | | | | | | | | | | | | | | | | | | Air Quality & Energy |
| Housing | | | | | | | | | | | | | | | | | | Urban Spaces & Tourism |
| Transport | | | | | | | | | | | | | | | | | | Safety |
| Transport | | | | | | | | | | | | | | | | | | Water Use |
| Transport | | | | | | | | | | | | | | | | | | Waste Management |
| Transport | | | | | | | | | | | | | | | | | | Air Quality & Energy |
| Transport | | | | | | | | | | | | | | | | | | Urban Spaces & Tourism |
| Safety | | | | | | | | | | | | | | | | | | Water Use |
| Safety | | | | | | | | | | | | | | | | | | Waste Management |
| Safety | | | | | | | | | | | | | | | | | | Air Quality & Energy |
| Safety | | | | | | | | | | | | | | | | | | Urban Spaces & Tourism |
| Water Use | | | | | | | | | | | | | | | | | | Waste Management |
| Water Use | | | | | | | | | | | | | | | | | | Air Quality & Energy |
| Water Use | | | | | | | | | | | | | | | | | | Urban Spaces & Tourism |
| Waste Management | | | | | | | | | | | | | | | | | | Air Quality & Energy |
| Waste Management | | | | | | | | | | | | | | | | | | Urban Spaces & Tourism |
| Air Quality & Energy | | | | | | | | | | | | | | | | | | Urban Spaces & Tourism |

**Figure A1.** AHP questionnaire for prioritizing the criteria. Note: 1 = Equal importance, 3 = Moderate importance, 5 = Strong importance, 7 = Very strong importance, 9 = Extreme importance (2,4,6,8- Values in-between).

**Question 2:** Between the Indicator **A** and **B**, which one is more important for the assessment of sustainable city development and management in Cambodia?

| A | 9 | 8 | 7 | 6 | 5 | 4 | 3 | 2 | 1 | 2 | 3 | 4 | 5 | 6 | 7 | 8 | 9 | B |
|---|---|---|---|---|---|---|---|---|---|---|---|---|---|---|---|---|---|---|
| **Demography** | | | | | | | | | | | | | | | | | | |
| Population density | | | | | | | | | | | | | | | | | | City's migration |
| Population density | | | | | | | | | | | | | | | | | | Household income |
| City's migration | | | | | | | | | | | | | | | | | | Household income |
| **Employment** | | | | | | | | | | | | | | | | | | |
| Labor forces | | | | | | | | | | | | | | | | | | Unemployment |
| Labor forces | | | | | | | | | | | | | | | | | | New jobs creation |
| Unemployment | | | | | | | | | | | | | | | | | | New jobs creation |
| **Housing** | | | | | | | | | | | | | | | | | | |
| Slum population | | | | | | | | | | | | | | | | | | Low-income housing |
| Slum population | | | | | | | | | | | | | | | | | | Quality of buildings |
| Low-income housing | | | | | | | | | | | | | | | | | | Quality of buildings |

**Figure A2.** AHP questionnaire for prioritizing assessment indicators of Demography, Employment, and Housing. Note: 1 = Equal importance, 3 = Moderate importance, 5 = Strong importance, 7 = Very strong importance, 9 = Extreme importance (2,4,6,8- Values in-between).

**Question 2:** *Cont.*

| A | 9 | 8 | 7 | 6 | 5 | 4 | 3 | 2 | 1 | 2 | 3 | 4 | 5 | 6 | 7 | 8 | 9 | B |
|---|---|---|---|---|---|---|---|---|---|---|---|---|---|---|---|---|---|---|
| **Transport** | | | | | | | | | | | | | | | | | | |
| Transport means | | | | | | | | | | | | | | | | | | Sidewalks |
| Transport means | | | | | | | | | | | | | | | | | | Parking lots |
| Transport means | | | | | | | | | | | | | | | | | | Traffic congestion |
| Sidewalks | | | | | | | | | | | | | | | | | | Parking lots |
| Sidewalks | | | | | | | | | | | | | | | | | | Traffic congestion |
| Parking lots | | | | | | | | | | | | | | | | | | Traffic congestion |
| **Safety** | | | | | | | | | | | | | | | | | | |
| Crimes prevention | | | | | | | | | | | | | | | | | | Construction safety |
| Crimes prevention | | | | | | | | | | | | | | | | | | Disasters prevention |
| Crimes prevention | | | | | | | | | | | | | | | | | | Insurances |
| Construction safety | | | | | | | | | | | | | | | | | | Disasters prevention |
| Construction safety | | | | | | | | | | | | | | | | | | Insurances |
| Disasters prevention | | | | | | | | | | | | | | | | | | Insurances |
| **Water Use** | | | | | | | | | | | | | | | | | | |
| Water supply | | | | | | | | | | | | | | | | | | Water consumption |
| Water supply | | | | | | | | | | | | | | | | | | Water reservoirs |
| Water consumption | | | | | | | | | | | | | | | | | | Water reservoirs |

**Figure A3.** AHP questionnaire for prioritizing assessment indicators of Transport, Safety, and Water Use. Note: 1 = Equal importance, 3 = Moderate importance, 5 = Strong importance, 7 = Very strong importance, 9 = Extreme importance (2,4,6,8- Values in-between).

**Question 2:** *Cont.*

| A | 9 | 8 | 7 | 6 | 5 | 4 | 3 | 2 | 1 | 2 | 3 | 4 | 5 | 6 | 7 | 8 | 9 | B |
|---|---|---|---|---|---|---|---|---|---|---|---|---|---|---|---|---|---|---|
| **Waste Management** | | | | | | | | | | | | | | | | | | |
| Solid waste collection | | | | | | | | | | | | | | | | | | Wastewater treatment |
| Solid waste collection | | | | | | | | | | | | | | | | | | Waste reduction |
| Wastewater treatment | | | | | | | | | | | | | | | | | | Waste reduction |
| **Air Quality and Energy** | | | | | | | | | | | | | | | | | | |
| Fine dust levels | | | | | | | | | | | | | | | | | | Urban forests |
| Fine dust levels | | | | | | | | | | | | | | | | | | Energy consumption |
| Fine dust levels | | | | | | | | | | | | | | | | | | Renewable energy |
| Urban forests | | | | | | | | | | | | | | | | | | Energy consumption |
| Urban forests | | | | | | | | | | | | | | | | | | Renewable energy |
| Energy consumption | | | | | | | | | | | | | | | | | | Renewable energy |
| **Urban Spaces and Tourism** | | | | | | | | | | | | | | | | | | |
| Urban parks | | | | | | | | | | | | | | | | | | Botanic gardens |
| Urban parks | | | | | | | | | | | | | | | | | | Heritage conservation |
| Urban parks | | | | | | | | | | | | | | | | | | Tourism growth |
| Urban parks | | | | | | | | | | | | | | | | | | Playgrounds |
| Botanic gardens | | | | | | | | | | | | | | | | | | Heritage conservation |
| Botanic gardens | | | | | | | | | | | | | | | | | | Tourism growth |
| Botanic gardens | | | | | | | | | | | | | | | | | | Playgrounds |
| Heritage conservation | | | | | | | | | | | | | | | | | | Tourism growth |
| Heritage conservation | | | | | | | | | | | | | | | | | | Playgrounds |
| Tourism growth | | | | | | | | | | | | | | | | | | Playgrounds |

**Figure A4.** AHP questionnaire for prioritizing assessment indicators of Waste Management, Air Quality and Energy, and Urban Spaces and Tourism. Note: 1 = Equal importance, 3 = Moderate importance, 5 = Strong importance, 7 = Very strong importance, 9 = Extreme importance (2,4,6,8- Values in-between).

**Appendix C. A Consistent Sample Calculated by AHP-OS Program**

*Appendix C.1. Criteria*

**Priorities**

These are the resulting weights for the criteria based on your pairwise comparisons:

| Cat | | Priority | Rank | (+) | (-) |
|---|---|---|---|---|---|
| 1 | Demography | 7.9% | 5 | 2.8% | 2.8% |
| 2 | Employment | 11.7% | 4 | 4.1% | 4.1% |
| 3 | Housing | 20.8% | 1 | 5.5% | 5.5% |
| 4 | Transport | 6.3% | 7 | 1.0% | 1.0% |
| 5 | Safety | 19.9% | 2 | 7.2% | 7.2% |
| 6 | Water Use | 15.9% | 3 | 4.0% | 4.0% |
| 7 | Waste Management | 6.8% | 6 | 2.5% | 2.5% |
| 8 | Air Quality & Energy | 6.0% | 8 | 2.9% | 2.9% |
| 9 | Urban Space & Tourism | 4.6% | 9 | 1.4% | 1.4% |

**Decision Matrix**

The resulting weights are based on the principal eigenvector of the decision matrix:

| | 1 | 2 | 3 | 4 | 5 | 6 | 7 | 8 | 9 |
|---|---|---|---|---|---|---|---|---|---|
| 1 | 1 | 0.50 | 0.33 | 1.00 | 0.25 | 0.50 | 2.00 | 2.00 | 2.00 |
| 2 | 2.00 | 1 | 1.00 | 2.00 | 0.33 | 0.50 | 2.00 | 2.00 | 2.00 |
| 3 | 3.00 | 1.00 | 1 | 3.00 | 1.00 | 2.00 | 4.00 | 4.00 | 4.00 |
| 4 | 1.00 | 0.50 | 0.33 | 1 | 0.33 | 0.50 | 1.00 | 1.00 | 1.00 |
| 5 | 4.00 | 3.00 | 1.00 | 3.00 | 1 | 1.00 | 2.00 | 3.00 | 3.00 |
| 6 | 2.00 | 2.00 | 0.50 | 2.00 | 1.00 | 1 | 3.00 | 3.00 | 3.00 |
| 7 | 0.50 | 0.50 | 0.25 | 1.00 | 0.50 | 0.33 | 1 | 2.00 | 2.00 |
| 8 | 0.50 | 0.50 | 0.25 | 1.00 | 0.33 | 0.33 | 0.50 | 1 | 3.00 |
| 9 | 0.50 | 0.50 | 0.25 | 1.00 | 0.33 | 0.33 | 0.50 | 0.33 | 1 |

Number of comparisons = 36
**Consistency Ratio CR** = 3.7%

Principal eigen value = 9.428
Eigenvector solution: 5 iterations, delta = 5.2E-9

*Appendix C.2. Indicators*

### Appendix C.2.1. Demography

**Priorities**

These are the resulting weights for the criteria based on your pairwise comparisons:

| Cat | | Priority | Rank | (+) | (-) |
|---|---|---|---|---|---|
| 1 | Population density | 24.0% | 2 | 3.2% | 3.2% |
| 2 | City's migration | 55.0% | 1 | 7.4% | 7.4% |
| 3 | Household income | 21.0% | 3 | 2.8% | 2.8% |

Number of comparisons = 3
**Consistency Ratio CR** = 1.9%

**Decision Matrix**

The resulting weights are based on the principal eigenvector of the decision matrix:

| | 1 | 2 | 3 |
|---|---|---|---|
| 1 | 1 | 0.50 | 1.00 |
| 2 | 2.00 | 1 | 3.00 |
| 3 | 1.00 | 0.33 | 1 |

Principal eigen value = 3.018
Eigenvector solution: 3 iterations, delta = 1.7E-8

### Appendix C.2.2. Employment

**Priorities**

These are the resulting weights for the criteria based on your pairwise comparisons:

| Cat | | Priority | Rank | (+) | (-) |
|---|---|---|---|---|---|
| 1 | Labor forces | 49.3% | 1 | 11.3% | 11.3% |
| 2 | Unemployment | 31.1% | 2 | 7.1% | 7.1% |
| 3 | New jobs creation | 19.6% | 3 | 4.5% | 4.5% |

Number of comparisons = 3
**Consistency Ratio CR** = 5.6%

**Decision Matrix**

The resulting weights are based on the principal eigenvector of the decision matrix:

| | 1 | 2 | 3 |
|---|---|---|---|
| 1 | 1 | 2.00 | 2.00 |
| 2 | 0.50 | 1 | 2.00 |
| 3 | 0.50 | 0.50 | 1 |

Principal eigen value = 3.054
Eigenvector solution: 4 iterations, delta = 2.1E-8

### Appendix C.2.3. Housing

**Priorities**

These are the resulting weights for the criteria based on your pairwise comparisons:

| Cat | | Priority | Rank | (+) | (-) |
|---|---|---|---|---|---|
| 1 | Slum population | 38.7% | 2 | 5.2% | 5.2% |
| 2 | Low-income housing | 44.3% | 1 | 6.0% | 6.0% |
| 3 | Quality of buildings | 16.9% | 3 | 2.3% | 2.3% |

Number of comparisons = 3
**Consistency Ratio CR** = 1.9%

**Decision Matrix**

The resulting weights are based on the principal eigenvector of the decision matrix:

| | 1 | 2 | 3 |
|---|---|---|---|
| 1 | 1 | 1.00 | 2.00 |
| 2 | 1.00 | 1 | 3.00 |
| 3 | 0.50 | 0.33 | 1 |

Principal eigen value = 3.018
Eigenvector solution: 4 iterations, delta = 1.9E-10

Appendix C.2.4. Transport

## Priorities

These are the resulting weights for the criteria based on your pairwise comparisons:

| Cat | | Priority | Rank | (+) | (-) |
|---|---|---|---|---|---|
| 1 | Transport means | 23.9% | 2 | 6.4% | 6.4% |
| 2 | Sidewalks | 19.8% | 3 | 2.9% | 2.9% |
| 3 | Parking lots | 16.8% | 4 | 3.6% | 3.6% |
| 4 | Traffic congestion | 39.5% | 1 | 5.7% | 5.7% |

Number of comparisons = 6
**Consistency Ratio CR** = 2.2%

## Decision Matrix

The resulting weights are based on the principal eigenvector of the decision matrix:

| | 1 | 2 | 3 | 4 |
|---|---|---|---|---|
| 1 | 1 | 1.00 | 2.00 | 0.50 |
| 2 | 1.00 | 1 | 1.00 | 0.50 |
| 3 | 0.50 | 1.00 | 1 | 0.50 |
| 4 | 2.00 | 2.00 | 2.00 | 1 |

Principal eigen value = 4.061
Eigenvector solution: 4 iterations, delta = 2.7E-9

Appendix C.2.5. Safety

## Priorities

These are the resulting weights for the criteria based on your pairwise comparisons:

| Cat | | Priority | Rank | (+) | (-) |
|---|---|---|---|---|---|
| 1 | Crimes prevention | 28.9% | 2 | 8.2% | 8.2% |
| 2 | Construction safety | 40.9% | 1 | 11.6% | 11.6% |
| 3 | Disasters prevention | 20.5% | 3 | 5.8% | 5.8% |
| 4 | Insurances | 9.6% | 4 | 2.7% | 2.7% |

Number of comparisons = 6
**Consistency Ratio CR** = 4.4%

## Decision Matrix

The resulting weights are based on the principal eigenvector of the decision matrix:

| | 1 | 2 | 3 | 4 |
|---|---|---|---|---|
| 1 | 1 | 0.50 | 2.00 | 3.00 |
| 2 | 2.00 | 1 | 2.00 | 3.00 |
| 3 | 0.50 | 0.50 | 1 | 3.00 |
| 4 | 0.33 | 0.33 | 0.33 | 1 |

Principal eigen value = 4.121
Eigenvector solution: 5 iterations, delta = 1.8E-8

Appendix C.2.6. Water Use

## Priorities

These are the resulting weights for the criteria based on your pairwise comparisons:

| Cat | | Priority | Rank | (+) | (-) |
|---|---|---|---|---|---|
| 1 | Water supply | 41.3% | 1 | 9.5% | 9.5% |
| 2 | Water consumption | 26.0% | 3 | 6.0% | 6.0% |
| 3 | Water reservoirs | 32.7% | 2 | 7.5% | 7.5% |

Number of comparisons = 3
**Consistency Ratio CR** = 5.6%

## Decision Matrix

The resulting weights are based on the principal eigenvector of the decision matrix:

| | 1 | 2 | 3 |
|---|---|---|---|
| 1 | 1 | 2.00 | 1.00 |
| 2 | 0.50 | 1 | 1.00 |
| 3 | 1.00 | 1.00 | 1 |

Principal eigen value = 3.054
Eigenvector solution: 4 iterations, delta = 9.2E-9

## Appendix C.2.7. Waste Management

### Priorities

These are the resulting weights for the criteria based on your pairwise comparisons:

| Cat | | Priority | Rank | (+) | (-) |
|---|---|---|---|---|---|
| 1 | Solid waste collection | 54.0% | 1 | 5.2% | 5.2% |
| 2 | Wastewater treatment | 29.7% | 2 | 2.8% | 2.8% |
| 3 | Waste reduction | 16.3% | 3 | 1.6% | 1.6% |

Number of comparisons = 3
**Consistency Ratio CR** = 1.0%

### Decision Matrix

The resulting weights are based on the principal eigenvector of the decision matrix:

| | 1 | 2 | 3 |
|---|---|---|---|
| 1 | 1 | 2.00 | 3.00 |
| 2 | 0.50 | 1 | 2.00 |
| 3 | 0.33 | 0.50 | 1 |

Principal eigen value = 3.009
Eigenvector solution: 3 iterations, delta = 9.9E-9

## Appendix C.2.8. Air Quality and Energy

### Priorities

These are the resulting weights for the criteria based on your pairwise comparisons:

| Cat | | Priority | Rank | (+) | (-) |
|---|---|---|---|---|---|
| 1 | Fine dust levels | 32.0% | 2 | 4.9% | 4.9% |
| 2 | Urban forests | 14.4% | 3 | 1.2% | 1.2% |
| 3 | Energy consumption | 39.2% | 1 | 5.2% | 5.2% |
| 4 | Renewable energy | 14.4% | 3 | 1.2% | 1.2% |

Number of comparisons = 6
**Consistency Ratio CR** = 0.8%

### Decision Matrix

The resulting weights are based on the principal eigenvector of the decision matrix:

| | 1 | 2 | 3 | 4 |
|---|---|---|---|---|
| 1 | 1 | 2.00 | 1.00 | 2.00 |
| 2 | 0.50 | 1 | 0.33 | 1.00 |
| 3 | 1.00 | 3.00 | 1 | 3.00 |
| 4 | 0.50 | 1.00 | 0.33 | 1 |

Principal eigen value = 4.021
Eigenvector solution: 4 iterations, delta = 1.4E-10

## Appendix C.2.9. Urban Spaces and Tourism

### Priorities

These are the resulting weights for the criteria based on your pairwise comparisons:

| Cat | | Priority | Rank | (+) | (-) |
|---|---|---|---|---|---|
| 1 | Urban parks | 30.4% | 1 | 7.0% | 7.0% |
| 2 | Botanic gardens | 10.2% | 5 | 1.2% | 1.2% |
| 3 | Heritage conservation | 26.2% | 2 | 4.0% | 4.0% |
| 4 | Tourism growth | 21.2% | 3 | 4.1% | 4.1% |
| 5 | Playgrounds | 12.1% | 4 | 2.0% | 2.0% |

Number of comparisons = 10
**Consistency Ratio CR** = 1.4%

### Decision Matrix

The resulting weights are based on the principal eigenvector of the decision matrix:

| | 1 | 2 | 3 | 4 | 5 |
|---|---|---|---|---|---|
| 1 | 1 | 3.00 | 1.00 | 2.00 | 2.00 |
| 2 | 0.33 | 1 | 0.33 | 0.50 | 1.00 |
| 3 | 1.00 | 3.00 | 1 | 1.00 | 2.00 |
| 4 | 0.50 | 2.00 | 1.00 | 1 | 2.00 |
| 5 | 0.50 | 1.00 | 0.50 | 0.50 | 1 |

Principal eigen value = 5.062
Eigenvector solution: 3 iterations, delta = 5.1E-8

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
