# Peer review of "Prioritizing Sustainable City Indicators for Cambodia"

_urbansci, doi:10.3390/urbansci3040104_

Round 1

Reviewer 1 Report

The paper is the second step of a previous published research but, if the reader doesn't know this last, it's impossible understand the meaning of the paper. The abstract doesn't present the background of the paper (motivations, target,..). The impression is to read the second chapter (or paragraph) of a book not a paper.

A scientific paper has to contain all the elements to understand the problem, the target, and how the author's choice solve all of that. All the scientific papers adds (or aims to add) a new piece of knowledge (in this case the previous research), but they are write in a way to recall all the information and scientific background.

 In this paper, there is not:

a description of the motivation of the research; a scientific background (literature review) of the different methodologies and techniques useful to reach the target, that is weight and prioritize the indicators.

The paper just follows the suggestion given by the same authors (and that is very strange) in their first article of this research to use the AHP technique. In the authors words: "Yet, this first-step research suggests the next-step research to prioritize these indicators by using the Analytic Hierarchy Process (AHP) technique..:".

Summarising, the paper wrote in this way is just an application of a technique, nothing more.

Nothing to say about the application of the AHP technique. 

Author Response

Dear Reviewer,

We would like to thank you for your time and efforts in reviewing our manuscript “Prioritizing Sustainable City Indicators for Cambodia” and providing thoughtful feedback.

We revised the manuscript accordingly and would like to respond to your comments as follows.

The paper is the second step of a previous published research but, if the reader doesn't know this last, it's impossible understand the meaning of the paper. The abstract doesn't present the background of the paper (motivations, target,…). The impression is to read the second chapter (or paragraph) of a book not a paper.

Response: We agreed with you. We accordingly rewrite the abstract by expressing in the beginning that this research is based on the previous research, and as suggested by the Reviewer 2, we included the background of the three rounds of Delphi methods and its results in the Abstract as well. Also, we further expressed the problem statement and objectives of the research in the rewrote Abstract.

A scientific paper has to contain all the elements to understand the problem, the target, and how the author's choice solve all of that. All the scientific papers adds (or aims to add) a new piece of knowledge (in this case the previous research), but they are write in a way to recall all the information and scientific background.

Response: Based on your comments, we tried to clearly present the problem statement and objectives of the research, including a hypothesis (Lines 83-96). With this improvement, we further discussed the results of this research (AHP) and previous research (Delphi) in the last part of the Discussion (Lines 619-627) and included in the Summary and Conclusions (Lines 678-694). In the Background, we tried to show the five major sources of indicators because the developed consensus indicators were started from and based on these indicators. And the attached Appendix can show readers what the reviewed indicators of these indicators are, and readers can also see how the developed consensus indicators are different from the reviewed indicators.

In this paper, there is not:

a description of the motivation of the research; a scientific background (literature review) of the different methodologies and techniques useful to reach the target, that is weight and prioritize the indicators. The paper just follows the suggestion given by the same authors (and that is very strange) in their first article of this research to use the AHP technique. In the authors words: "Yet, this first-step research suggests the next-step research to prioritize these indicators by using the Analytic Hierarchy Process (AHP) technique…".

Response: As mentioned earlier, we present the motivation of the research in Lines 83-96. We agreed that only following the previous suggestion, this research used AHP is not an appropriate way. In the previous version, we did not explain well why this research chose AHP for this prioritization; we just follow the previous research’s suggestions because we thought that we had discussed this technique with the Delphi in previous research. However, this issue should be presented again as a scientific background like your suggestions. Therefore, we improve it accordingly (Lines 83-96).

Summarising, the paper wrote in this way is just an application of a technique, nothing more. Nothing to say about the application of the AHP technique. 

Response: This research was carried out to prioritize the consensus indicators developed by the previous research. As mentioned in the limitation of this research, the AHP technique is very new to Cambodia. Therefore, presenting the application of the AHP method in prioritizing sustainable city indicators for Cambodia in this research can introduce the AHP technique to the future relevant prioritization projects and research in Cambodia. With this application, this research further aims to show that the levels of importance of the consensus indicators verified by the Delphi method in previous research cannot use for prioritizing or ranking the consensus indicators (Lines 619-627 in the Discussion) and (Lines 678-684 in the Summary and Conclusion).

We thank you for the opportunity to improve our manuscript.            

Best regards,

The Authors

Reviewer 2 Report

This research is very interesting and well elaborated. Nevertheless, there is a previous research -corresponding to reference [1]- and there is a confusion about what was produced by the first research and what by the second one. If I understand well, it is the second one that is presented in this manuscript.

Furthermore, in the summary, it is not clear which is the first-step research and which is the next-step research. In general, the abstract must me improved. Please rewrite the abstract trying to express clearly a.the fact that your actual research is based on a previous one b. the research design procedure, with the 3 rounds of Delphi method and finally the results of the surveys. 

I am also recommending an attentive proof-reading in English to make your text easily comprehensible by every reader.  

Author Response

Dear Reviewer,

We would like to thank you for your time and efforts in reviewing our manuscript “Prioritizing Sustainable City Indicators for Cambodia” and providing thoughtful feedback.

We revised the manuscript accordingly and would like to respond to your comments as follows.

This research is very interesting and well elaborated. Nevertheless, there is a previous research -corresponding to reference [1]- and there is a confusion about what was produced by the first research and what by the second one. If I understand well, it is the second one that is presented in this manuscript.

Response: Based on your comments, we understand that the confusion was because we used the subjects “First research and second research, including first-step research and second-step research”. Therefore, we accordingly changed these subjects to “previous research and this research”. Yes, the second one is presented in this research and the previous one is presented only in the Introduction and last part of the Discussion.

Furthermore, in the summary, it is not clear which is the first-step research and which is the next-step research.

Response: By using the subjects “previous research and this research” instead of the subjects “first-step research and next-step research”, we improved all the unclear parts, especially the summary and conclusion parts of the manuscripts.

In general, the abstract must me improved. Please rewrite the abstract trying to express clearly a. the fact that your actual research is based on a previous one b. the research design procedure, with the 3 rounds of Delphi method and finally the results of the surveys. 

Response: We agreed with your suggestions. We accordingly rewrote the abstract by expressing in the beginning that this research is based on the previous research, the research design procedure with the three rounds of Delphi, and the results of the surveys. 

I am also recommending an attentive proof-reading in English to make your text easily comprehensible by every reader.

Response: We double checked the English language and style in our manuscript and also received help from a foreign professor to help check the English style. However, there will be full proof-reading if the manuscript is accepted.

We thank you for the opportunity to improve our manuscript.            

Best regards,

The Authors

Round 2

Reviewer 1 Report

General comment

this version is more clear than the previous one even if there is not a critical review of the methods and techniques useful to prioritize the indicators and what is the advantage to use the AHP instead of the others.

Detailed comment:

line 11: modify "...the previos research.." with "..a previuos research..."  line 19: modify " ..can use..." with "... can be used..." line 20: delete the question mark line 37: modify "...the previos research.." with "..a previuos research..."  line 46: modify "...(To.." with "...(to.." line 47: modify "...(To.." with "...(to.." line 84: modify "..And as" with "...As" line 85: delete "...continuously..." line 86-89: the sentence is not clear line 91-94: the sentence is not clear line 94: do you mean "the hypothesis of this research is to verify if the priority or rank.... is different from the ....."?

Author Response

Dear Reviewer,

We would like to thank you again for your time and efforts in reviewing our manuscript and providing thoughtful feedback. We revised the manuscript accordingly and would like to respond to your comments as follows:

General comment

this version is more clear than the previous one even if there is not a critical review of the methods and techniques useful to prioritize the indicators and what is the advantage to use AHP instead of others.

Detailed comment:

line 11: modify "...the previos research.." with "..a previuos research..."  line 19: modify " ..can use..." with "... can be used..." line 20: delete the question mark line 37: modify "...the previos research.." with "..a previuos research..."  line 46: modify "...(To.." with "...(to.." line 47: modify "...(To.." with "...(to.." line 84: modify "..And as" with "...As" line 85: delete "...continuously..."

Response: We modified and deleted the points you suggested.

line 86-89: the sentence is not clear line 91-94: the sentence is not clear. line 94: do you mean "the hypothesis of this research is to verify if the priority or rank.... is different from the ....."?

Response: We rewrote the not-clear sentences (Lines 86-96) by orderly explaining the problem statement, objectives, and research questions. We also rewrote the hypothesis, we mean “AHP’s rank is different from Delphi’s rank” (Lines 95-96) and as we discussed in the discussion (Lines 619-627).

We thank you again for the opportunity to improve our manuscript.

Best regards,

The Authors